# Gut Microbiome Modulation Based on Probiotic Application for Anti-Obesity: A Review on Efficacy and Validation

**DOI:** 10.3390/microorganisms7100456

**Published:** 2019-10-16

**Authors:** Kaliyan Barathikannan, Ramachandran Chelliah, Momna Rubab, Eric Banan-Mwine Daliri, Fazle Elahi, Dong-Hwan Kim, Paul Agastian, Seong-Yoon Oh, Deog Hwan Oh

**Affiliations:** 1Department of Food Science and Biotechnology, College of Agriculture and Life Sciences, Kangwon National University, Chuncheon, Gangwon-do 24341, Korea; bkannanbio@gmail.com (K.B.); ramachandran865@gmail.com (R.C.); rubab.momna@gmail.com (M.R.); ericdaliri@yahoo.com (E.B.-M.D.); elahidr@gmail.com (F.E.); 2Kangwon Institute of Inclusive Technology, Kangwon National University, Chuncheon, Gangwon-do 24341, Korea; donghwan@kangwon.ac.kr; 3Department of Plant Biology and Biotechnology, Loyola College, Chennai 600-034, India; agastian@loyolacollege.edu; 4Three & Four Co., Ltd., 992-15, Jusan-ri, Hojeo-myeon, Wonju-si 26460, Korea; 3n4world@hanmail.net

**Keywords:** obesity, gut microbiome, probiotics, mechanism, diet

## Abstract

The growing prevalence of obesity has become an important problem worldwide as obesity has several health risks. Notably, factors such as excessive food consumption, a sedentary way of life, high sugar consumption, a fat-rich diet, and a certain genetic profile may lead to obesity. The present review brings together recent advances regarding the significance of interventions involving intestinal gut bacteria and host metabolic phenotypes. We assess important biological molecular mechanisms underlying the impact of gut microbiota on hosts including bile salt metabolism, short-chain fatty acids, and metabolic endotoxemia. Some previous studies have shown a link between microbiota and obesity, and associated disease reports have been documented. Thus, this review focuses on obesity and gut microbiota interactions and further develops the mechanism of the gut microbiome approach related to human obesity. Specifically, we highlight several alternative diet treatments including dietary changes and supplementation with probiotics. The future direction or comparative significance of fecal transplantation, synbiotics, and metabolomics as an approach to the modulation of intestinal microbes is also discussed.

## 1. Introduction

Obesity is a significant health problem that is associated with many disorders including hypertension, diabetes, and dyslipidemia. Worldwide, more than 1900 million adults (over the age of 18) were clinically observed to be obese in 2016 [1]. According to the World Health Organization (WHO) [1], for adults, body mass index (BMI) > 25 is normal or equal, while BMI > 30 is considered obese. Obese people have a lifespan that is seven years shorter than that of non-obese people [2]. The prevalence of obesity has increased in recent years due to environmental and genetic factors that probably interact. Global environmental causes of obesity include lower rates of physical activity, excessive food consumption, consumption of calorie-rich and preserved food, and the use of some drugs. In addition, obesity is linked to the development of urological diseases such as erectile dysfunction and benign prostatic hyperplasia, which may affect health [3].

The Food and Agriculture Organization (FAO) and WHO define microbiota as “live microorganisms which provide hosts with health benefits, when administered adequately” [4]. Every human has about 1 to 2 kg of microbiota, which comprises 10 trillion microorganisms including a minimum of 1000 different species [5]. These microbes are important for the permeability of the gastrointestinal mucosa and host immune systems. Most of the gut flora include bacteria that represent 60% of the dry fecal mass. Fungi and protozoa are also part of the intestinal flora, but their activities are not well known [6]. In the food industry, there are several widespread probiotic strains such as *L. casei*, *L. johnsonii*, *B. lactis*, *L. rhamnosus*, *S. cerevisiae*, and *B. animals*, which modulate the immune response [7,8]. *Lactobacillus gasseri* BNR17 was shown to inhibit adiponectin and leptin secretions, and decrease adipose tissue and plasma [9]. Other probiotic microbes such as *B. longum*, *L. acidophilus*, *P. acidilactici*, and *L. casei* also have hypocholesterolemic effects [10]. The *L. casei* strain was shown to lower oxidative stress [11] and suppress CD4+ T cell effectors by lowering the concentration of pro-inflammatory molecules [12] through antioxidant, immune modulatory, and antidiabetic effects.

Previous studies have discussed possible treatments for obese patients based on translational medicine. Although obesity is not a disease, it is considered to be a deficiency, according to Larsen et al. [13]. This deficiency can be rectified based on the theorized link between the composition of intestinal microbial and metabolic conditions [14], although previous studies on obesity have provided no definitive therapy. Hence, recent studies have focused on modulating beneficial microbiota through supplementation with probiotics, which are expected to play a significant role in the neutralization of this deficiency [14]. A similar improvement has been observed in autoimmune diseases such as type 1 diabetes [7,15,16,17]. The altered microbiota also causes the invasion of opportunistic pathogens, which are resistant to oxidative stress and capable of decreasing the concentration of sulfates and impeding bacterial butyrate growth [18]. In this context, enzymes are involved in glucose homeostasis, which causes neutralization of the disorder due to peripheral insulin resistance or insufficient insulin in the B cells [19]. Hamad et al. [20] found that the addition of *L. gasseri* SBT2055 reduced mesenteric adipose tissue mass, adipocyte size, and serum leptin levels in lean Zucker rats. Serum and hepatic cholesterol reductions were observed in combination with improved fecal fatty acid and total neutral steroid excretion in obese or lean Zucker rats. The observed properties of triacylglycerols, phospholipids, and cholesterol were attributed to reduced absorption of the maximum lymphatic content. Following the use of probiotic drugs, total cholesterol, triacylglycerol, lipoprotein of high-density and cholesterol of low-density lipoprotein levels were not altered [21,22]. However, the blood glucose level was reduced both quickly and post-radially due to increased glucose sensitivity [23]. Likewise, the *L. casei* Shirota strain also increased the insulin sensitivity and decreased the glucose intolerance of mice with diet-induced obesity. *Lactobacillus* fermentation, produced by multiple bacteria from wheat and barley flowers, decreased the acquired body weight, prerenal and epididymal fat, and total serum cholesterol of animals with diets containing the highest amount of fat [24]. Multiple strains of Bifidobacteria supplements generated important reductions in weight gain and blood sugar and leptin concentrations without important changes in the weight of the fat pad in rats [15,25]. A summary of previous assessments of the properties of probiotic experimental models and clinical studies [22] on obesity is provided in Table 1 and Table 2. Additionally, research has shown that obesity is associated with increased *Firmicutes*, while the concentration of *Bacteroidetes* is decreased in obese patients [24,26]. Specifically, patients with obesity experienced a significant reduction in *Bacteroidetes*, which increased the relative concentration of *Firmicutes*, which is positively correlated with plasma glucose concentration [27]. Current studies are focused on prebiotics such as arabinoxylan and arabinoxylan oligosaccharides, which have shown positive outcomes throughout the control of adiposity-connected metabolic disorders. This review, which supports the role of probiotics in the management of obesity, provides an overview of previous trial studies.

## 2. Dysbiosis in Gut Microbiota

In human physiology, the continuous role of gut microbes ties our health to balanced gut microbiota. Unstable microbiota or dysbiosis of the intestines is correlated with underlying inflammatory bowel disease conditions, for example, *Clostridium difficile* [23], obesity, and autoimmune disorders [23,41,42]. The prevalence of microbiota dysbiosis in obese patients has been discussed in numerous studies [43,44]. In addition to *Bacteroidetes* and *Firmicutes*, important human commensal phyla and several bacterial taxonomic groups (families, genera, and even species) have been described to be linked by obesity dysbiosis. Some bacterial families including *Enterobacteriaceae* [45] and *Prevotellaceae* [41] have been associated with obesity at the familial level, but *Christensenellaceae* [41] has been found to be enriched in subjects with low BMI.

*Lactobacillus* was found to be increased in adolescents after a weight-loss program and was extremely prevalent in overweight or obese children [46,47,48,49]. *Bacteroides* were prominent among obese people and their abundance was positively linked with BMI [49]. The *Roseburia* genus appeared to benefit obesity because of the increased feces when obese people consumed more indigestible polysaccharides [50,51,52]. An adverse correlation between BMI and *Bifidobacterium* was shown [34]. Likewise, *Erwinia*, *Succinivibrio* [53], *Alistipes* [54], and *Oscillospira* [55] species were found to be more abundant in normal-weight subjects than in obese patients [56,57,58,59,60]. *Coprococcus catus*, *Blautia hydrogenotrophic*, *Ruminococcus bromii*, *Ruminococcus obeum*, and *Eubacterium ventriosum* were significantly correlated with obesity in Japanese people [60,61,62], whereas the presence of *Bacteroides thetaiotaomicron*, *Bacteroides faecichinchillae*, *Flavonifractor plautii*, *Blautiawexlerae*, and *Clostridium bolteae* was associated with lean people [63]. Moreover, the gut commensal bacterium *Akkermansia muciniphila* has been used as an aid in the treatment of obesity [63]. Additionally, the enormous variety of microbial biology contributes to differential weight gain and obesity growth at the species level as well as the strain level. Microbiome studies therefore focus on mechanical understanding, and the correlative association of intestinal microbes with obesity is gaining traction [64,65].

## 3. Mechanistic Studies Linked to Obesity Metabolism 

Some mechanisms for the function of gut microbiota in the progression of obesity have been suggested, as shown in Figure 1.

### 3.1. Bile Acid Metabolism

Bile acids actively help to resolve dietary fat uptake in the small intestine. Glycine or taurine is conjugated with cholesterol, which is synthesized in the liver. In the intestine, bacterial deconjugation and dihydroxylation transform primary bile acid into secondary bile acid [66]. In addition to the gastrointestinal mechanism, bile acids have been considered to inhibit growth by disturbing the membrane reliability of various gut bacteria [67] such as the probiotics *Bifidobacteria* and *Lactobacilli* [68]. Bile acids are also known to be ligands for nuclear farnesoid X-receiving receptor (FXR). A review of the gut microbiome of FXR-null mice and wild-type mice showed that microbiota promote dietary obesity through FXR signals [69]. Regarding G Protein-Coupled Bile Acid Receptor 1 (GPBAR1)/Takeda G protein-coupling receptor-5 (TGR5), activation caused exclusively by bile acids was also described for glucose homeostasis by releasing gastric glucagon-like peptide-1 [70]. Subsequent studies have also reported the role of TGR5 in the anti-obesity effects of bariatric surgery. Fecal microbiota transplantation in FXR-null mice fed with a high-fat diet resulted in lower weight gain in germ-free mice than the transplantation of broad-type counterparts, showing that the gut microbiota mediates dietary obesity [71,72].

### 3.2. Short-Chain Fatty Acids

In the anaerobic distal gut intestinal tract, bacteria are used as microbial growth substrates for ingestible polysaccharides (i.e., fiber) [73]. Short-chain fatty acids (SCFAs) such as acetate, propionate, and butyrate are the primary yield. The human contribution of these SCFAs is estimated to be around 80–200 kcal/day, which is eventually consumed by various organisms [74]. In a previous study, decreased fecal mass of bacteria that produce butyrate in obese patients was associated with decreased consumption of dietary carbohydrates (e.g., polysaccharides, resistant starch, and vegetable oligosaccharides) [75]. A study with a similar design showed significantly decreased concentrations of fecal butyrate, total SCFAs, and *Bifidobacterium* in patients with low-fiber diets [76]. Low-fiber diets have been shown to increase pathogen sensitivity by maintaining the gastric mucosa microbiota in the host. Microbial fermentation SCFAs have been shown to reduce the gallbladder pH, modifying microbiota by producing a niche that encourages an increase in butyrate-producing bacteria. Additionally, concerning the existence of an energy source and the colonic pH-modulating function, SCFAs are signaling molecules for at least two G-protein-coupled receptors, Gpr41 and Gpr [77]. Hence, a high-fiber diet can aid in the management of obesity through intestinal modulation and downstream pathway production caused by SCFAs.

### 3.3. Metabolic Endotoxemia

Obesity has long been defined by chronic inflammation and resistance to insulin [78]. Some reports have indicated that intestinal permeability and metabolic endotoxemia increase throughout the development of obesity. The relationship between obesity and chronic inflammation was elusive until Cani et al. [79] found a high-fat dietary elevation of plasma lipopolysaccharide (LPS), or metabolic endopotoxemia. In contrast, Cani et al. [79] also defined metabolic endotoxemia as a chronically high plasma LPS disorder at 10–50 times less than the septic conditions of LPS. Metabolic endotoxemia (inducing pro-inflammatory and oxidative stress) was observed in obese mice (C57bl6/J) that consumed normal chow (composed of ground wheat, corn, or oats, alfalfa and soybean meal, protein with minerals and vitamins), but it might be induced in normal mice through an obesogenic (high-fat) diet [56]. In a comparison, the proportional abundance of *Firmicutes* was significantly lower in low-fat diet (LFD) than high-fat diet (HFD) fed mice, but Proteobacteria were not significantly changed between the two groups [80]. In these studies, dietary increases of endotoxin were linked to enhanced fat deposit, systemic and tissue-specific inflammation, and resistance to insulin [79,81]. In healthy people, metabolic endotoxemia was proven to reduce insulin sensitivity by 35%, which was correlated with the consumption of energy and may occur as a consequence of consuming high-fat or high-carbohydrate foods [81]. The complementary results covering cross-sectional, impending, longitudinal, and experimental studies emphasize the clinical significance of metabolic endotoxemia in inflammatory obesity and cardiometabolic abnormalities.

### 3.4. Probiotic Effects on Plasma Lipids

In recent decades, inflammation, innate immune mechanisms, and metabolic pathways have been shown to be controlled by lipids (obtained through the diet) [82]. Dietary lipids have an unequivocal nutritional significance, but they also have a function in (pro-inflammatory) interactions of nuclear receptors [83]. This contains the families of peroxisome proliferator active receptors (PPARs) and liver X receptors (LXRs), which are pivotal in inflammatory and metabolic pathways. Many fatty acids trigger all three PPAR family members, thus improving insulin action and eliminating the development of pro-inflammatory cytokines such as TNF-α [84,85]. All dietary metabolites and lipids activate g-protein receptors (Gprs). For example, by activating Gpr43 via dietary metabolite acetate, lipolysis in adipocytes was found to directly decrease, which is key to lowering plasma-free fatty acid levels. This shows a potential therapeutic role for Gpr43 in regulating lipid metabolism [86,87]. Backhed et al. [88] found that microbial suppression of the intestinal factor adipocyte induced by fasting promoted adiposity, suggesting that increased Fiaf expression and activity may lead to leanness. It is evident that the pathophysiology of atherosclerosis is linked to changes in the interpersonal intestinal microbiome [89]. Foods rich in lipid phosphatidylcholine, referred to as lecithin such as eggs, milk, fish, liver, poultry, and red meat are a significant source of dietary choline [90]. Bile acids are also important regulators of lipid and cholesterol metabolism, which facilitates intestinal absorption and the transport of nutrients and vitamins. As the production of bile takes place in the liver, 95% of bile acids are absorbed in the terminal ileum and are then reabsorbed by the liver, according to enterohepatic circulation. As a result, lipid catabolism absorbs the acids. The intestinal microbiota converts primary bile salts into secondary bile salts through bile acid dehydroxylation [66,91]. While a low dose of oral antibiotics can influence intestinal microbiota composition and human bile acids, it has also been shown to have various effects on glucose metabolism [92,93]. In murine models, bile acids have been shown to promote the release of GLP-1 (enteroendocrine L-cell), resulting in the activation of Takeda G protein-coupling receptor-5 (TGR5). However, some studies have failed to find high levels of plasma bile acid for metabolic purposes during the first days of bariatric surgery [94].

### 3.5. Microbiota and Obesity: From the Intestines to the Brain

Desire, food consumption, and energy balance are essential components of a complex neuroendocrine factor and receptor network that mediates the pathways between the gastrointestinal tract and the brain [95]. After ingestion of a meal, the stimulus of the nutrients moving through the digestive tract causes complex neural and hormonal signals that inform the brain. Due to the effects of these neurotransmitters, the gut microbiota is considered a second brain. It is becoming increasingly evident that dopamine, epinephrine, norepinephrine, gamma-aminobutyric acid, serotonin, indole metabolites, and some other gut microbiota receptors impact dietary biases. This signaling, mediated by afferent nerve fibers of the autonomous nervous system such as the vagus nerve, sends information from the intestine to the nucleus tractus solitarius (NTS) and effector fibers, which are projected into the smooth muscles of the gut. The NTS information, distributed to the hypothalamus, regulates the energy balance, appetite, and dietary intake of the arcuate-core (ARC) neurons. The ARC provides orexigenic neuro Y family peptides, associated with both agouti and anorexigenic peptides, and regulates transcripts of proopiomelanocortin [12,88]. The other peripheral gastrointestinal peptides including pancreatic polypeptide and peptide tyrosine-tyrosine, reduce appetite. Neuromedin U and neuromedin S suppress feeding, while melanin-concentrating hormone, orexin A, and orexin B stimulate food intake. In addition, the Y family neuropeptides are mainly produced and released by specialized enteroendocrine cells, which accumulate in the gastrointestinal (GI) wall [96,97].

Metabolites obtained from microbiota such as short-chain fatty acids can bind to enteroendocrine cell receptors and alter the release of enteric hormones into the systemic circulation. In animal and human studies, the fermentation of nondigestible carbohydrates by intestinal microbiota has been demonstrated to increase the production of SCFAs and the excretion of gut hormones [98]. Acetate, which is secreted from gut microbes, is the main SCFA [99]. However, an increased concentration of acetate due to altered microbiota causes parasympathetic nervous system activation, supporting the spread of glucose, ghrelin secretion, and obesity [97,100]. Another study on germ-free mice showed an upregulated response in the intestinal sweet signaling protein T1R2, which resulted in a higher intake of carbohydrates [101]. Some of the lactic acid bacteria can convert glutamate into gamma-aminobutyric acid (GABA) to express GABA-binding proteins [102,103] and serotonin [95,97,104,105,106], which act as synaptic neurotransmitter signals that regulate appetite. 

Serotonin mediates the effect of melanocortins on the appetite by modulating the appetite-suppression effects [102]. In another study of germ-free mice, the intestinal sweet signaling protein T1R3 led to increased consumption of sweet nutrient solutions [64,68,103,107]. Modulation of melanocortin neurons, which control bodyweight homeostasis, was shown to reduce appetite [108,109]. GABA, the central nervous system′s principal inhibitor, stimulates feeding and synaptic discharge from vague protein-expressing neurons in the hypothalamic ARC, providing standard energy balance control (Figure 2) [108,110,111,112]. In this context, it is necessary to clarify the mechanisms that connect the gut–brain axis and human microbiota to provide a potential approach for managing obesity and its associated metabolic disorders.

## 4. Obesity Associated with Other Factors

### 4.1. Effect of Probiotics in Obesity-Associated Kidney Patients

Obesity increases the possibility of chronic kidney disease (CKD) and progression to end-stage renal disease (ESRD). Probiotics could reduce the production of some uremic toxins. The protein metabolite hypothesis, proposed by Niwa et al. [109], states that organic anion transporters in the renal tubules take on toxins generated by protein putrefaction by gut microbes. For example, *Lactobacillus murinus* supplementation prevents development of hypertension from high salt intake in hypertensive mice [113,114]. Ichii et al. [115] showed that LPS in mouse podocytes displayed a pro-inflammatory phenotype, reduced podocyte-specific gene expression, and reduced cell viability. In the case of ESRD patients, approximately 190 microbial operational taxonomic units were considered to be distinctly in abundance when compared with healthy checks [116]. In CKD patients, lower numbers of *Prevotellaceae* and *Lactobacillaceae* families (both regarded to be natural colonic microbiota) and 100 times greater *Enterococci* and *Enterobacteria* species were identified [116]. Likewise, an oral supplement given to MLR (mixed lymphocyte reaction) mice with spontaneous lupus nephritis enhanced kidney function and general survival by decreasing the intestinal permeability and systemic inflammation of the symbiont strains of *Lactobacillus* (*L. gasseri*, *L. rhamnosus*, *L. oris*, *L. reuteri*, and *L. johnsonii*). Another study found an enhanced gastrointestinal barrier, reduced internal swelling, and accumulation of uremic toxins in spontaneous 5/6 hypertensive nephrectomized rats (a typical CKD model) treated with *Lactobacillus acidophilus*, resulting in reduced kidney damage [117]. Treatment with various bacteria lowered lipid peroxidation and improved antioxidant enzymes such as superoxide dismutase (SOD) and catalase (CAT) in acetaminophen-induced uremic rats and chrome-induced oxidative stress rats [118,119,120]. Additionally, reduced oxidative stress was correlated with reduced renal necrosis [119,121,122].

In summary, probiotics can protect against renal injury and dysfunction by reducing swelling, apoptosis, and oxidative stress. Some findings might be inconsistent because of the distinction between bacterial species and the patient population or animal models used during the research. Indeed, obesity changing the intestinal environment through bacterial composition can be considered as a cause of renal injury. Notable findings indicate that particular personal and microbiome characteristics allow specific glucose response predictions, which can result in customized microbiome modulation through dietary, prebiotic, and probiotic intervention [123]. Modulating the gut′s bacterial balance by using probiotics could be a suitable choice in cases of obesity to improve kidney function.

### 4.2. Effects of Prebiotics and Dietary Fiber on Obesity Treatment

The FAO and WHO describe prebiotics as nondigestible food products that benefit the host by selectively promoting the development and activity of one or more of the existing colon bacterial species, thus improving host health [124]. This perception usually includes indigestible, nonhydrolyzable forms of carbohydrate (e.g., fructo-oligosaccharides (FOSs), galacto-oligosaccharides (GOSs), soybean oligosaccharides, cyclodextrins, inulin, gluco-oligosaccharides, xylo-oligosaccharides, lactulose, lacto-sucrose, and isomalto-oligosaccharides), which have the potential to reach distal parts of the human gastrointestinal tract [125]. There is increasing evidence that prebiotic therapy positively influences the composition of gut microbes, enhancing the growth of *Lactobacillus* and *Bifidobacterium* in the gastrointestinal tracts of obese animals [126]. For instance, human breast milk is a rich source of milk oligosaccharides (prebiotics candidate), which boost the development of beneficial bacteria (*Bacteroides* and *Bifidobacterium*) and inhibit the adherence of pathogens such as *E. coli*, *Campylobacter jejuni*, and *Helicobacter pylori* [127]. Some trials have shown that these modifications in oligofructose-treated obese and diabetic mice [128], as much as prebiotic carbohydrate-treated ob/ob mice, have to do with accelerated enteroendocrine cell growth, glucose homeostasis, and leptin sensitivity [129]. These changes were also related to the high production of intracellular glucagon-like peptide-2 (GLP-2), the intestinotrophic pro-glucagon-derived peptide associated with gut permeability, thus decreasing obesity-related gastric inflammatory and cardiovascular disorders. In this context, Everard et al. [128] reported on non-obese metabolic phenotypes described by decreased triglyceride concentrations, adipose tissue, and muscle lipid infiltration in oligofructose-treated animals. There is wide proof of the positive impacts of *L. rhamnosus* GG therapy in the pediatric population regarding obesity-related nonalcoholic fatty liver disease (NAFLD). In most clinical research, the hepatic accumulation of TAG and/or cholesterol in liver tissue, specified as steatosis, can be reduced by prebiotics. This impact might be interesting, as 25% to 75% of obese people have NAFLD. Although the energy intake function is a concern, studies have not shown any effects of long-term prebiotic supplementation in pre-meal inulin and galacto-oligosaccharides or short-term fructose-oligosaccharide [130] treatment; others showed that oligofructose or inulin intake in both non-obese and obese individuals decreased their overall energy intake for at least two weeks [130,131,132,133]. Several findings on prebiotics showed significant beneficial outcomes on body weight, waist circumference, BMI, lipid profile, fat deposition, and chronic inflammation status, which may lead to alternative approaches in the management and treatment of obesity and associated metabolic disorders [123].

## 5. Correlation of Obesity with Immune System and Transmission through Next Generation

### 5.1. Influence of High-Fat Diet in Obesity Patients on Correlated Immunosenescence

Increased intestinal mucosal absorption (e.g., leaky gut syndrome), especially in relation to immune modifications, can lead to important harm to the GI tract, causing bacteria, toxins, nutrients, and waste to leak from the bowels into the bloodstream, with a severe autoimmune response, particularly if such toxins deplete the liver [134]. In a mouse model, Moya-Pérez et al. [135] found that the gut ecosystem with *B. pseudocatenulatum* CECT 7765 modulated HFD-induced infiltration of immune cells and intestinal and peripheral inflammation, with respective improvements in obesity-related metabolic dysfunction. The study also showed that the anti-inflammatory effects of these bifidobacterial strains involve both inborn and adaptive immune functions involving B lymphocytes. Zhang et al. [136] showed two strains of *Lactobacillus rheuteri* that were piglet-insulated as ZJ617, with strong adhesive capability, and ZJ615, with low adhesive capability. Likewise, they analyzed the immunomodulatory effects of strains with diverse adhesive abilities. Kowalska et al. [137] first outlined, and numerous trials still show, that leptin is elevated under the impact of HFD relative to control diets in rats [138,139,140]. It has been proven in rats fed the same diet that HFD increases leptin content [141]. In rats, leptin might be reduced with HFD within 72 h regardless of increased body weight [142]. The reasons for the immunosenescence phenomenon have not yet been fully explained. In addition, there have been several confounders such as body fat mass, which separately influence the immune response, but can differ significantly among heterogeneous aging people. Elevated body fat mass was suggested to not have the same adverse effect on elderly people as young people. This concluded with research showing a survival benefit for geriatric populations above the age of 65 when BMI reaches 25 kg/m^2^, generally identified as overweight [143,144]. In addition to fat tissue and integrated immune cells, mediators are known to be immediately involved such as adipocytes or several mostly pro-inflammatory cytokines [144,145,146], which migrate to the immune function. Nevertheless, the dietary intake of probiotics is widely considered to be beneficial for health, mainly due to its immunomodulatory properties in obesity treatment.

### 5.2. Effect of Maternal Obesity on Newborns 

In recent decades, the incidence of several prenatal and early postnatal variables connected with the growth of childhood adiposity (such as prematurity and low birth weight [147,148,149], gestational diabetes [150], surplus body mass gain during gestation, and formula feeding) has also continued in recent decades, in conjunction with an enhanced prevalence of childhood obesity. Interestingly, the incidence of these perinatal risk factors has risen more acutely in developing countries [151,152,153]. Gestational diabetes mellitus (GDM) increases the risk of disproportionate adiposity and macrosomia in infants. Long-term GDM is correlated with a greater danger of child obesity and baby metabolism [154,155]. Boyle et al. [156] showed that parental obesity increases the capability of mesenchymal umbilical cord stems for abiogenesis, which leads to infant adiposity. Childhood obesity is of significant concern, given the elevated danger that obese children will become obese adults and thus develop severe comorbidities such as metabolic syndrome, diabetes, and cardiovascular disorders [157]. The gut microbiome is acknowledged as an environmental factor in recent innovations in gnotobiotic mouse sequencing technology, which influences the metabolism of the entire organism by influencing the energy equilibrium along with regulatory signals of peripheral and essential food intake, inflammation, and intestinal block functions, thus stimulating body weight. Gut microbiota are a specific entity in the body that has its own genome with a gene pool bigger than its host. A novel relationship between obesity and diabetes has been ascribed to the extensive physiological functions of intestinal gut microbiota in extra-intestinal tissues such as adipose tissue [158,159]. Thus, it plays a significant role in obesity and diabetes disease pathology. Studies with germ-free mice that showed protection from dietary-induced obesity (DIO) growth first showed the underlying processes of the microbiota attributable to host metabolism [154,157,159]. Notable animal studies concluded that probiotics during pregnancy and lactation reduced maternal HFD-induced dietary programming associated with maternal obesity, which suggests that modifying the parental gut microbiota might be a helpful approach to enhance paternal and offspring metabolic findings [160]. Human studies by Vähämiko et al. [161] revealed that probiotic supplements throughout pregnancy may impact the DNA methylation status by some obesity promoters and weight gain genes in both mothers and their offspring. Everard et al. [162] showed that *L. rhamnosus* GG started one month before birth and continued until six months after birth altered children′s development patterns by reducing unjustified weight gain during infancy. A promising prevention and therapeutic approach for GDM may be the use of probiotic supplementation. Microbiota dysbiosis was treated with probiotics supplementation, and probiotics have emerged as an efficient intervention to improve the health of preterm infants. 

### 5.3. Modulation of Gut Microbiota Based on Controlled Diets

Diet is the most important health factor in the prevention of obesity and is also closely associated with modulation in microbiota. Various studies have shown that susceptibility to genetic obesity can be linked with an obesogenic environment (e.g., a major change in diet that influences the gut microbiota, physical inactivity, and a sedentary lifestyle) in determining an obesity epidemic [163]. There are currently many common diets including standard and Western diets. Probiotics are often investigated for their impact on human health, and extensive research is carried out on new strains with probiotic potential. In this section, diets supplemented with probiotics designed to reduce the weight of obese patients are shown to significantly alter the species composition of gut microbiota.

### 5.4. Gut Microbiome Based on a Normal Diet

Diet plays an important role in the composition of gut flora [164] and is linked with three clusters of gut microbiota profiles, which are considered to be enterotypes [165]. Arumugam et al. [165] stated that various distinct microbial gut profile types, dominated by *Prevotella*, *Bacteroides*, and *Ruminococcus*, are not limited to geographic origin [165]. The impact of a specific diet has a specific influence on enterotype change such that *Bacteroides* are associated with a rich protein and animal fat diet, while *Prevotella* is associated with high carbohydrate consumption (Figure 3). Zimmer et al. [166] stated that omnivores have greater bacterial richness than vegetarians. The results showed that the proportions of *Escherichia coli* and *Bacteroides*, *Bifidobacterium*, and *Enterobacteriaceae* species were suggestively lower in samples from vegans than those from omnivores. The proportions of *E. coli* biovars and *Enterobacteriaceae*, *Klebsiella*, *Enterobacter*, *Citrobacter*, and *Clostridium* did not vary among the groups. These microbes in vegetarians were proportional to those in vegans and omnivores.

### 5.5. Gut Microbiome Based on the Western Diet

In many countries around the world, ongoing Westernization, urbanization, and mechanization processes have resulted in the adoption of a sedentary lifestyle and high-fat, high-energy diets [100,167,168]. Fat is a major part of the diet and is a substrate for the production of intestinal microbes and short-chain fatty acids [169,170]. A high-fat diet rich in both fish oil (ω-3 polyunsaturated fatty acids) and pig fat (lard) (HL diet, mainly saturated fatty acids) has a stronger impact on microbiota modulation, but pig fat has a stronger impact on *Bacteroidetes* and *Bilophila* enhancement, while fish oil increases *Actinobacteria* [171,172]. Numerous studies have reported that obesity influences the levels of specific phyla, with reported changes in the ratios of *Firmicutes* and *Bacteroidetes* in both humans and rodents. In lean individuals, the differences in microbiota were correlated with energy losses through the stool [173]. As the collected energy of 150 kcal was increased by 20%, there was a corresponding decrease in *Bacteroidetes*. It has been reported that the relative proportion of *Bacteroidetes* is lower in obese patients than in lean people [174].

It has been suggested that the increased abundance of *Firmicutes* in the intestinal microbiota of obese patients improves energy harvesting from the Western diet and thus promotes better caloric absorption and subsequent weight gain [170]. Moreover, in genetically or diet-induced obese mice and rats, the proportion of *Firmicutes* to *Bacteroidetes* enhancement has been reported in relation to controls in HFD [107,175]. Recent evidence shows that the acute addition of fat to the Western diet leads to modulation of intestinal response signals, resulting in the modulation of energy intake, lipid accumulation, and inflammation [176]. The Western diet is associated with increased gut *Firmicutes* and a simultaneous increase in the lipid profile, reducing the variety of intestinal flora (Figure 4).

### 5.6. Gut Microbiome Based on a Diet Supplemented with Probiotics

Many previous studies were based on the anti-obesity effects of probiotic supplementation, specifically on the reduction of lipogenesis, inflammation, and weight loss (Figure 5). In particular, the probiotic *L. rhamnosus* GG strain has been used in obesity studies [177,178]. *L. rhamnosus* GG treatment replaced HFD by reducing adiposity in high-fat-diet-fed mice through the enhancement of adiponectin production, which protects animals from insulin resistance and can reduce liver adiposity [179]. In addition, purified exopolysaccharides can reduce the risk of adipogenesis in *L. rhamnosus* GG cells as well as fat pads and inflammation by the expression of Toll-like receptor 2 in HFD-fed mice [178]. Wistar rats showed a significant reduction in total body and visceral adipose tissue weight and improved insulin sensitivity after a short treatment with a mixture containing 14 probiotic biomass bacteria of the genera *Bifidobacterium*, *Lactococcus*, and *Propionibacterium* [180]. A clinical health study involving 49 overweight and obese adults showed an essential relationship between the abundance of *A. muciniphila* and metabolic health. Indeed, the healthiest metabolic conditions were particularly present for people with enhanced gene richness and *A. muciniphilia* throughout the fasting plasma, triglyceride, and body fat distribution [181]. Probiotic treatment of nonviable *A. muciniphila* cells killed through pasteurization was recently demonstrated to increase their ability to reduce fat mass development, insulin resistance, and dyslipidemia [182,183]. This appeared to be due to a certain protein interaction between Toll-like receptor 2 and the process of pasteurization, which is located in the cell wall of *A. muciniphila* [182].

In this context, the reversal of endotoxemia and fluidized adipose of *A. muciniphila*, which is typically decreased, suggests potential probiotic applications in the treatment of obesity [160,162,181]. A simultaneous metagenomics analysis of the gut microbiota showed that 67 paths including those involved in fat and protein metabolism, carbohydrate catabolism, and the biosynthesis of gut microbiota are affected by dietary intervention, suggesting that dietary activity has a strong effect on the metabolic activity of gut microbiota.

## 6. Future Direction of Anti-Obesity Treatment

### 6.1. Fecal Microbiota Transplantation

Fecal microbiota transplantation (FMT), which involves the transfer of fecal microbiota from specific donors to recipients, is an easy way to manipulate the intestinal microbiota. In many cutting-edge microbiota investigations, the recipients are typically germ-free mice. This method is currently being used clinically to treat difficult diseases such as *Clostridium difficile*, as it is an innovative, effective, and safe approach [184]. As the composition of the microbiota is complex and unrefined, FMT is in accordance with the International Scientific Association for Probiotics and Prebiotics (ISAPP) guidelines [181]. Recently, several studies have assessed the possibility of modulating the intestinal microbiota using FMT for the treatment of other diseases including obesity and metabolic disorders. Gough et al. [185] stated that FMT had a positive effect on 317 *C. difficile*-infected patients in 27 case studies. In contrast, it was shown that 92 % of *Clostridium difficile* infection (CDI) cases were eliminated through fecal microbiota transplantation (FMT) [181]. Currently, eight registered trials on FMT for the treatment of obesity are taking place [186]. 

### 6.2. Combined Effects of a Synbiotic-Modulated Diet

Synbiotics can have a more significant impact on intestinal and host health than the isolated consumption of pre- or probiotics, given that they supply probiotic bacteria together with a prebiotic component that stimulates probiotic bacteria survival and gastrointestinal growth [187]. Evidence indicates that synbiotics can be useful for changing the microbiota composition [188]. In a 12-week study, Roller et al. [189] investigated the combination of special oligofructose-enriched inulins (SYN1), *Lactobacillus rhammnosus* GG, and *Bifidobacterium lactis* Bb12, and found that they caused deceases of 16%, 18%, and 31%, respectively, in *Clostridium perfringens*. In vitro studies have shown that synbiotics are more efficient at modulating gut microflora than prebiotics and probiotics [190].

### 6.3. Correlation of Metagenomics and Metabolomics Approaches for Obesity Treatment

Studies on intestinal microbial metabolomes undoubtedly have considerable potential to aid in determining the metabolic pathway of gut microbiota and microbiota–host interactions [176,191]. However, some barriers prevent researchers from determining the production and regulation of these modifications in intestinal metabolomics and the particular intestinal bacteria that are responsible for their differences. Metatranscriptomic and metagenomic analyses indicate additional tools to obtain a helpful view on gut microbiota [192]. Recently, Sridharan et al. [193] developed a genome annotation data-based metabolic network model using mass spectrometry (MS) based metabolomics to detect 26 metagenomics metabolites. The most commonly used metabolic profiling methods are MS [81,82] and proton nuclear magnetic resonance (1H NMR) spectroscopy [80]. 1H NMR produces reproducible and robust metabolomic data on biofluids (blood, cell media, urine, etc.) and needs minimum sample preparation. In contrast, MS is more fragile, and is able to detect metabolites at significantly reduced concentrations. To enhance the resolution, MS is generally combined with either liquid or gas chromatography [83]. Incorporating metabolome in other omics data such as metaproteomics, metagenomics, and metatranscriptomics will deepen our understanding of complex biosynthetic tracts of gut microbiota. Such integrative studies can also contribute to the development of noninvasive diagnostic tools for obesity, optimize personalized medicines, and improve the effectiveness of individual dietary supplements. A number of problems in gut metabolomics still need to be addressed in the future.

## 7. Conclusions

The task of this review was to investigate the body of literature on the use of probiotics for obesity treatment. Probiotics have shown weight reduction characteristics in animal research and human studies, and several factors have been suggested to explain the anti-obesity effects. Previously, the role of microflora in metabolic regulation, disease, and genetic disorders was an unexplored area of gastroenterology. Currently, the role of microflora has been identified through molecular sequencing such as metagenomics and metabolomics, and its significant coregulation was studied extensively in bile salts, SCFAs, metabolic endotoxemia, and obesity. However, the specificity of probiotics makes it difficult to determine a specialized path for their operating mechanism. Human studies must be conducted to identify subtle differences in the levels of metagenomics and metabolomics to exploit the potential of probiotics for the treatment of obesity and its associated metabolic diseases. Preclinical models will hopefully lead to the quick detection of unique and cost-effective probiotic strains, which will require the completion of significant clinical studies over the next few years in order to determine whether they are suitable for human consumption. Restoration or modulation of the microbiota composition using “live bacteria (probiotics) or foods that cannot be fully digested such as prebiotics (oligosaccharides), or synbiotics, or even fecal transplants” is being targeted as a possible approach for the prevention and treatment of obesity. Another issue is that the optimal size of probiotics to be used for treatment is still unclear. Although numerous experimental models have been used, differences in administration such as resuspension solution differences in obese models and host animals might have influenced the results and conclusions of previous studies. However, this review indicates that progress in this direction may be useful for improving intervention strategies in the management of obesity and its associated metabolic deficiencies.

## Figures and Tables

**Figure 1 microorganisms-07-00456-f001:**
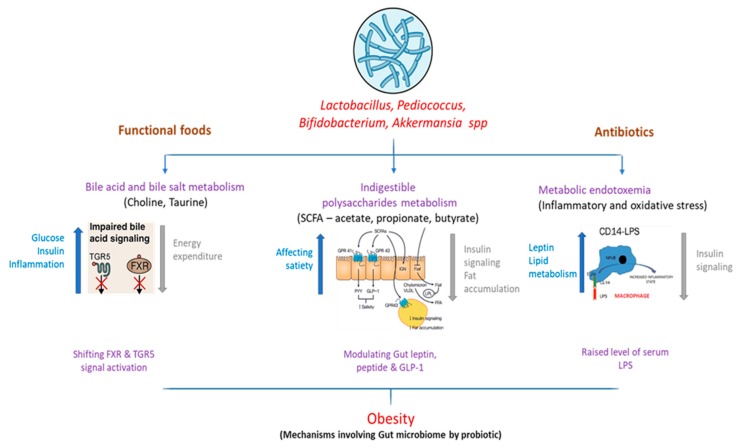
Interaction of altered gut microbiome and downstream metabolic effects influencing obesity.

**Figure 2 microorganisms-07-00456-f002:**
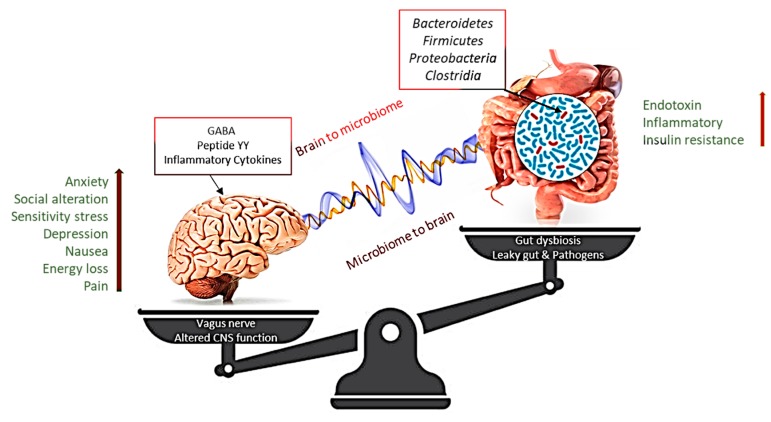
Altering microbiota show that intestinal dysbiosis can adversely affect human physiology, causing improper signaling of the intestinal brain axis and associated effects on central nervous system (CNS) functions, resulting in obesity. In contrast, stress at the CNS level can influence gut function and lead to microbiota disturbances. GABA, gamma-aminobutyric acid. The [Right arrow] indicate gradual increase in Endotoxin, Inflammatory and Insulin resistance, which is correlated with the enhanced level [Left arrow] of Anxiety, social alteration, Sensitivity stress Depression, Nausea, Energy loss and pain.

**Figure 3 microorganisms-07-00456-f003:**
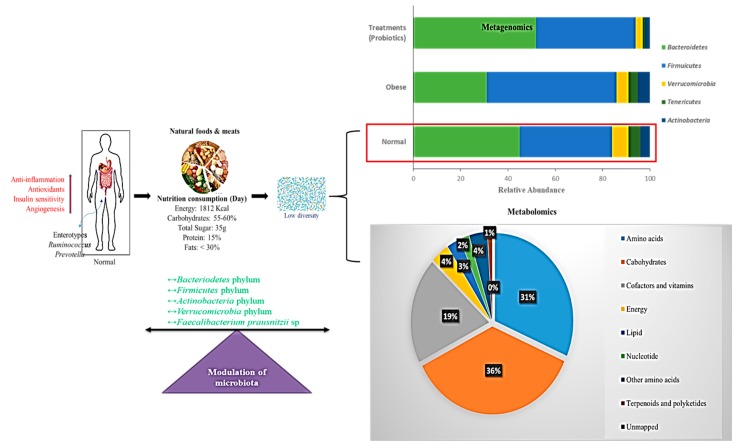
Schematic representation of a normal diet in the modulation of gut microbiota. Interactions between normal diet, nutrition, energy intake, and modulation of microbiota are shown. Bar chart represents the metagenomics results on operational taxonomic units (OTUs) grouped in phyla. Pie chart shows the percentage of metabolic effects influencing normal diet. Red arrows indicate increased activity

**Figure 4 microorganisms-07-00456-f004:**
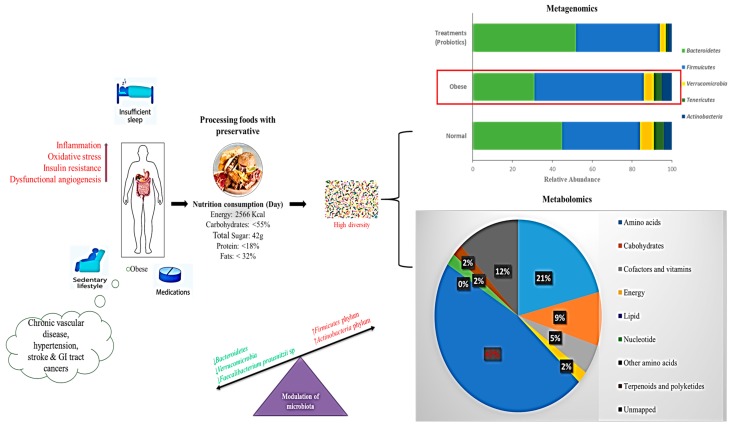
Schematic representation of a Western diet modulating gut microbiota showing interactions between nutrition, energy intake, and modulation of microbiota. Bar chart shows the metagenomics results of operational taxonomic units (OTUs) grouped in phyla. Pie chart shows the percentage of metabolic results influenced by a Western diet (>lipid level); Red arrows indicate increased activity.

**Figure 5 microorganisms-07-00456-f005:**
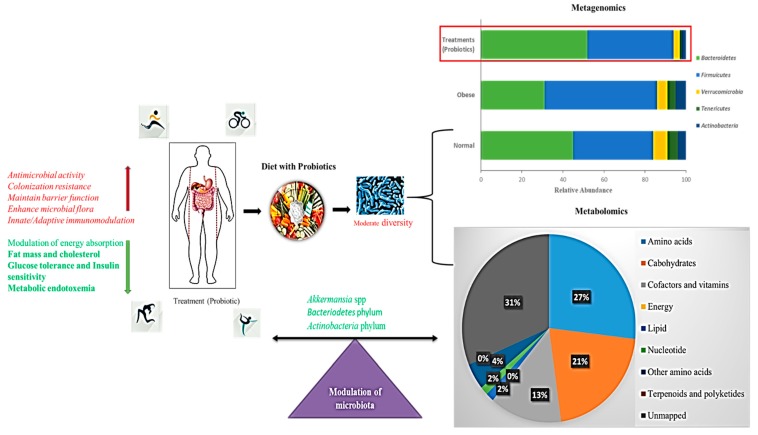
Schematic representation of a probiotic-supplemented diet modulating gut microbiota. Bar chart shows the metagenomics results of operational taxonomic units (OTUs) grouped in phyla. Pie chart shows the percentage of metabolic effects influencing a probiotic-supplemented Western diet (decreased lipid levels); Red arrows indicate increased activity, Green arrows indicate decreased activity.

**Table 1 microorganisms-07-00456-t001:** Probiotic strains with efficacy against obesity in in vivo animal models. Reprinted with permission from Mishra et al. [28]. Copyright year publisher.

Probiotic Strain	Dose Level	Experimental Study	Experimental Results	Reference
*P. Pentosaceus* LP28	3 × 10^9^ CFU (1.25 × 10^9^ CFU/g) for 6 weeks	Mice (C57BL/6Jcl) DIO (D12492) mice fed with high-fat diet	↓ Epididymal fat, liver cholesterol, liver TG	Zhao et al. [29]
*L. plantarum* FH185	1 × 10^9^ CFU for 6 weeks	Male C57BL/6 mice fed with high-fat diet	↑ *Lactobacillaceae* HFD-FH185, *Bacteroidaceae*, and *Porphyromonadaceae* increased slightly ↓ Adipocyte of epididymal fat pads	Park et al. [30]
*L. plantarum* HAC01	1 × 10^8^ CFU/day for 8 weeks	Male C57BL/6 mice fed with high-fat diet (HFD)	↑ Lipid oxidative gene expression, including acyl-coenzyme A oxidase (ACOX), carnitine, palmitoyltransferase1 (CPT1), peroxisome, proliferator-activated receptor gamma, coactivator 1-alpha (PGC-1α), and peroxisome proliferator-activated receptor ↓ Body weight by 12%, mesenteric adipose depot by 50%, and adipocyte of epididymal fat	Park et al. [31]
*L. plantarum* K21	1 × 10^9^ CFU/day for 8 weeks	High-fat-diet-induced C57/BL6J mouse	↑ Weight and epididymal fat accumulation (50% and 23%), gut permeability and improved fecal microbiota composition (increased *Lactobacillus* spp. and *Bifidobacterium* spp., reduced *Clostridium perfringens* ↑ Leptin, total cholesterol (TC), and triglycerides TG (56%, 13%, and 33%) ↓ Hepatic TC and TG (25% and 45%) and expression of hepatic PPARγ mRNA	Wu et al. [32]
*L. plantarum* LG42	1 × 10^9^ CFU/day for 12 weeks	C57BL/6 mice fed a high-fat diet (HFD)	↓ Body weight (38%), serum level, insulin (60%) and leptin (39%), PPARγ, aP2, C/EBPα, lipoprotein lipase (LPL), and liver X receptor α (LXRα)	Park et al. [33]

CFU, colony forming unit; PPAR, peroxisome proliferator active receptor; DIO, dietary-induced obesity. (↓ decreased, ↑ increased).

**Table 2 microorganisms-07-00456-t002:** Recent human trial for obesity study by probiotic strains. Reprinted with permission from Mishra et al. [28]. Copyright year publisher.

Strains	Dose Level	Experimental Study	Experimental Results	References
*L. rhamnosus* GG	1 × 10^9^ CFU/day for 20 weeks	Human	↔ Prevented GDM in overweight and obese pregnant women	Callaway et al. [34]
*L. rhamnosus* GG and *B. lactis* BB12	6.5 × 10^9^ CFU capsules/day for 12–17 weeks	Human	↑ Gestational weight gain or birthweight	Karaponi et al. [35]
*B. adolescentis* IVS-1 and *B. lactis* BB-12	1 × 10^9^ CFU/day for 3 weeks	Human	↑ Colonic permeability	Krumbeck et al. [36]
*L. gasseri* BNR17	1 × 10^9^ CFU/day for 12 weeks	Human	↑ Reduced visceral fat mass in obese adults	Kim et al. [37]
*B. breve B-3*	5 × 10^10^ CFU/day for 12 weeks	Human	↑ Improved HDL cholesterol↓ Reduced body fat	Minami et al. [38]
*L. gasseri* SBT2055	1 × 10^7^ CFU/day for 12 weeks	Human	↑ Increased fat emulsion droplet size ↓ Suppression of lipase-mediated fat hydrolysis	Ogawa et al. [39]
*L. casei* DN 114001	1 × 10^10^ CFU/day for 12 weeks	Human	↓ Cost-efficient reduction of prevalence of antibiotic-associated diarrhea	Dietrich et al. [40]

GDM, gestational diabetes mellitus; HDL, high-density lipoprotein. (↓ decreased, ↑ increased, ↔ moderate).

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
