# Peer review of "Gut Microbiome Modulation Based on Probiotic Application for Anti-Obesity: A Review on Efficacy and Validation"

_microorganisms, 2019, doi:10.3390/microorganisms7100456_

Round 1

Reviewer 1 Report

This manuscript entitled “Gut microbiome modulation based on probiotic application towards anti-obesity: a review on efficacy and validation” by Barathikannan K et al. is a review on the probiotic use in the management of obesity. Several errors were found, and some of them were serious. Several major questions could be raised.

In line 81, in what species (e.g. human), supplementation with Bifidobacteria caused significant decline in weight gain, while it was ineffective in rats? Please specify. In line 85, is the sentence correct? (obesity is associated with increased Bacteroidetes and decreased Firmicutes) In line 106, Bacteroidetes might probably be associated with loss of body weight rather than BMI. Please check. In line 113-4, do authors mean that weight gain and obesity development could be different depending on the types microbiota administered or microbiota composition of the host? The meaning is unclear. In line 127, bile acids have cholesterol ring and derived from cholesterol. However, bile acids are not cholesterol, glycine or taurine. In line 137, The transplantation of “cecal microbiota” --- could be correct. And in lines 138-139, the meaning is “gut microbiota promotes diet-induced obesity through FXR and FX contributes to increased adiposity by changing the gut microbiota” rather than “gut microbiota mediates dietary obesity. In line 165, “could be inducted into mackerel (B57bl6/j)” is hard to understand. In line 168, CD14 is not an LPS molecule. What does LPS CD14 mean? In Line 170, TLR4-NF-kB does not improve the transcription of proinflammatory cytokines. What do authors mean by “improve”? In line 173, it is not clear why “However” was inserted. In line 216, it is not clear what ‘followed by hosts” means. In line 235, “lard extracted as fish oil” is hard to understand. Lard is animal fat. In line 242, the sentence does not make much sense. Do authors mean that relative proportion of Bacteroidetes is lower in patients with less weight loss? In line 246, “to increase’ seems to be necessary between reported and in relation to. In line 262, do authors mean “protect animals from HFD-induced insulin resistance”? “may protect animals from resistance to HFD-induced insulin production” is hard to understand. In line 276, the reference 107 seems to be a wrong one. Please change to a correct one. In line 282, is there any reference showing adipose fluidization by Akkermansia? In line 303, does ‘intro-endocrine cells’ mean ‘enteroendocrine cells’? Still it is hard to understand why separation of enteroendocrine cells from gut peptides contributes to the transmission of signals. In line 314, is the reference 116 correct?   “Some of the bacteria such as GABA and serotonin” is nor correct. GABA and serotonin are not bacteria. In line 341, it is not clear what ‘investigated diseases were eliminated in 92%” means. In line 357, what does ‘metabolized pattern’ mean? Is pattern of metabolites more appropriate? In line 379, ‘questionable studies’ do not seem to be good words and can be insulting to the authors. Potential bad effects of “probiotics” need to be mentioned. Important papers regarding Akkermansia seem to be missing in the review (such as Shin N-R et al. An increase in the Akkermansia sp. population induced by metformin treatment improves glucose homeostasis in diet-induced obese mice. Gut 63:727-735, 2014; Depommier C et al. Nat Med 25:1096, 2019; Barcena C et al. Healthspan and lifespan extension by fecal microbiota transplantation into progeroid mice ---. Nat Med in press, 2019.

Author Response

Response to Reviewer 1 Comments

Manuscript ID: microorganisms-570580

Type of manuscript: Review

Title: Gut Microbiome Modulation based on Probiotic Application towards Anti-obesity: A Review on Efficacy and Validation

The authors were grateful for the editorial and reviewer’s valuable comments to improvise the manuscript quality towards reader. The manuscript was revised according to the valuable suggestion. All changes in the revised manuscript were highlighted in red font. The language of the research article were edited by a MDPI English editing service (english-10989) and proof read. A point-by-point response to comments is included below. We are grateful to the reviewer for providing valuable comments on the article and also for your precious time.

Comments and Suggestions for Authors

This manuscript entitled “Gut microbiome modulation based on probiotic application towards anti-obesity: a review on efficacy and validation” by Barathikannan K et al. is a review on the probiotic use in the management of obesity. Several errors were found, and some of them were serious. Several major questions could be raised.

We are thankful to the reviewer for providing valuable comments on the article and also for your precious time. The comments and suggestions raised have been addressed in the article. The response for individual comments is given below. As per the reviewers valuable suggestion the following comments were edited and specific line numbers were provided appropriately towards each individual comments.

In line 81, in what species (e.g. human), supplementation with Bifidobacteria caused significant decline in weight gain, while it was ineffective in rats? Please specify.

Changes were effected in the manuscript (Line: 81-84)

Multiple strains of Bifidobacteria supplements generated important reductions in weight gain and blood sugar and leptin concentrations without important changes in weight of the fat pad in rats.

References (18, 26)

In line 85, is the sentence correct? (obesity is associated with increased Bacteroidetes and decreased Firmicutes)

Changes were effected in the manuscript (Line: 88-87)

As per the reviewer suggestion the manuscript these sentences, it be modified and rephrased.

Additionally, research has shown that obesity is associated with increased Firmicutes, while the concentration of Bacteroidetes is decreased in obese patients (15, 16).

References (15, 26)

In line 106, Bacteroidetes might probably be associated with loss of body weight rather than BMI. Please check.

Changes were effected in the manuscript (Line: 105-109)

Bacteroides were prominent among obese people and their abundance was positively linked with BMI (48).The Roseburia genus appeared to benefit obesity because of the increase in feces when obese people consumed more indigestible polysaccharides (50). An adverse correlation between BMI and Bifidobacterium has been disclosed (39).

References (48, 50, 39)

In line 113-4, do authors mean that weight gain and obesity development could be different depending on the types microbiota administered or microbiota composition of the host? The meaning is unclear.

Changes were effected in the manuscript (Line: 115-118)

Also, the enormous variety of microbial biology contributes to differential weight gain and obesity growth roles at species level and also at strain level. Microbiome studies therefore focus on mechanical understanding and the correlative association of intestinal microbes with obesity are gaining traction.

References (64, 65).

In line 127, bile acids have cholesterol ring and derived from cholesterol. However, bile acids are not cholesterol, glycine or taurine.

Changes were effected in the manuscript (Line: 131-132)

Bile acids actively help to resolve dietary fat uptake in the small intestine. The glycine or taurine derived from cholesterol produced from in the Liver.

In line 137, The transplantation of “cecal microbiota” --- could be correct. And in lines 138-139, the meaning is “gut microbiota promotes diet-induced obesity through FXR and FX contributes to increased adiposity by changing the gut microbiota” rather than “gut microbiota mediates dietary obesity.

Changes were effected in the manuscript (Line: 141-143)

The transplantation of cecal microbiota Fxr-null mice fed with a high-fat diet resulted in a lower weight gain in germ-free mice than the transplantation of broad-type counterparts, which showed that the gut microbiota mediates dietary obesity.

References (71, 72)

In line 165, “could be inducted into mackerel (B57bl6/j)” is hard to understand.

Changes were effected in the manuscript (Line: 167-168)

Metabolic endotoxemia was observed in genetically obese mice (C57bl6/J) consume normal chow and mackerel based chow that consume the obesogenic diet.

In line 168, CD14 is not an LPS molecule. What does LPS CD14 mean?

Changes were effected in the manuscript (Line: 170-171)

In that line typographical error, removed from the line “CD14”

In Line 170, TLR4-NF-kB does not improve the transcription of proinflammatory cytokines. What do authors mean by “improve”?

Changes were effected in the manuscript (Line 170-174)

These LPS binding, CD14 gene, and their Toll-like co-receptor 4 (TLR4) activate the nuclear factor-kB (NF-kB) inflammatory pathway, influence on the transcription of several pro-inflammatory cytokines involved in obesity and other metabolic pathogeneses (80).

References (80)

In line 173, it is not clear why “However” was inserted.

Changes were effected in the manuscript (Line 176-179)

The complementary results covering cross-sectional, impending, longitudinal and experimental studies emphasize the clinical significance of metabolic endotoxemia in inflammatory obesity and cardiometabolic abnormalities.

In line 216, it is not clear what ‘followed by hosts” means.

Changes were effected in the manuscript (Line: 388-390)

The impact of specific diet have an specific influence on enterotype change such as Bacteroides are associated with a rich protein and animal fat diet, while Prevotella is associated with high carbohydrate consumption

In line 235, “lard extracted as fish oil” is hard to understand. Lard is animal fat.

Changes were effected in the manuscript (Line: 407-410)  

High-fat diets rich in both fish oil (ω3-PUFAs) and Pig (Lard) fat (HL diet, mainly saturated fatty acids) has a stronger impact on microbiota modulation in that Pig fat  induce stronger impact on Bacteroidetes and Bilophila enhancement, while Actinobacteria increased in fish oil rich food diet (169, 170).

Reference (Number 169, 170).

In line 242, the sentence does not make much sense. Do authors mean that relative proportion of Bacteroidetes is lower in patients with less weight loss?

Changes were effected in the manuscript (Line: 407-409)

High-fat diets rich in both fish oil (ω3-PUFAs) and Pig (Lard) fat (HL diet, mainly saturated fatty acids) has a stronger impact on microbiota modulation in that Pig fat  induce stronger impact on Bacteroidetes and Bilophila enhancement, while Actinobacteria increased in fish oil rich food diet (169, 170).

In line 246, “to increase’ seems to be necessary between reported and in relation to.

Changes were effected in the manuscript as follows Line (416-418)

Moreover, in genetically- or diet-induced obese mice and rats, the proportion of Firmicutes to Bacteroidetes enhancement has been reported in relation to controls in HFD (103, 164).

Reference (103, 164)

In line 262, do authors mean “protect animals from HFD-induced insulin resistance”? “may protect animals from resistance to HFD-induced insulin production” is hard to understand.

Changes were effected in the manuscript as follows Line (435-436)

rhamnosus GG can replace HFD, which protect animals from insulin resistance and can reduce liver adiposity.

Reference (173)

In line 276, the reference 107 seems to be a wrong one. Please change to a correct one.

Changes were effected in the manuscript as follows Line (172-174)

This appeared to be due to a certain protein interaction between the Toll-like receptor 2 and the process of pasteurization, which is located in the cell wall of A. muciniphila (175).

References (175)

Plovier, H.; Everard, A.; Druart, C.; Depommier, C.; Van Hul, M.; Geurts, L.; Chilloux, J.; Ottman, N.; Duparc, T.; Lichtenstein, L. A purified membrane protein from Akkermansia muciniphila or the pasteurized bacterium improves metabolism in obese and diabetic mice. Nat Med. 2017; 23(1):107–13.

In line 282, is there any reference showing adipose fluidization by Akkermansia?

Changes were effected in the manuscript as follows Line (455-457)

In this context, the reversal of endotoxemia and adipose fluidized of A. muciniphila, which is typically decreased, suggests potential probiotic applications in the treatment of obesity (159,160, 176)

Reference

159.Shin, N.R.; Lee, J.C.; Lee, H.Y.; Kim, M.S.; Whon, T.W.; Lee, M.S.; Bae, J.W. An increase in the Akkermansia spp. population induced by metformin treatment improves glucose homeostasis in diet-induced obese mice. Gut 2014; 63:727–35.

160.Everard, A.; Belzer, C.; Geurts, L.; Ouwerkerk, J.P.; Druart, C.; Bindels, L.B.; Guiot, Y.; Derrien, M.; Muccioli, G.G.; Delzenne, N.M.; de Vos, W.M.; Cani, P.D. Cross-talk between Akkermansia muciniphila and intestinal epithelium controls diet-induced obesity. Proc Natl Acad Sci U S A, 2013; 110:9066–

176.Plovier, H.; Everard, A.; Druart, C.; Depommier, C.; Van Hul, M.; Geurts, L.; Chilloux, J.; Ottman, N.; Duparc, T.; Lichtenstein, L. A purified membrane protein from Akkermansia muciniphila or the pasteurized bacterium improves metabolism in obese and diabetic mice. Nat Med. 2017; 23(1):107–13.

In line 303, does ‘intro-endocrine cells’ mean ‘enteroendocrine cells’? Still it is hard to understand why separation of enteroendocrine cells from gut peptides contributes to the transmission of signals.

Changes were effected in the manuscript as follows Line (218-224)

The ARC provides the orexigenic neuro Y family peptides, in associated with both agouti and anorexigenic peptides, and regulates transcripts of proopiomelanocortin (94, 12). The other peripheral gastrointestinal peptides including the pancreatic polypeptide and peptide tyrosine-tyrosine reduce appetite. Neuromedin U and neuromedin S suppress feeding, whilst melanin-concentrating hormone, orexin A, and orexin B stimulate food intake. In addition, the Y family neuropeptide were mainely produced and released by specialized enteroendocrine cells, which were accumilated in gastrointestinal wall (GI wall) (95, 96). 

References (12, 94, 95, 96)

In line 314, is the reference 116 correct?   “Some of the bacteria such as GABA and serotonin” is nor correct. GABA and serotonin are not bacteria.

Changes were effected in the manuscript as follows Line (234-236)

Some of the bacteria producing metabolite, such as gamma-aminubutyric acid (GABA) (102,108) and serotonin (96, 101, 109, 110, 111), are neuroactive signals that affect central appetite control.

References (96, 101, 102, 108, 109, 110, 111)

In line 341, it is not clear what ‘investigated diseases were eliminated in 92%” means.

Changes were effected in the manuscript as follows Line (472-473)

In contrast, it was shown that 92 % of cases of investigated diseases were eliminated, although several studies were combined with systemic approaches (129).

References

Cerdó, T.; García-Santos, J.A.; G Bermúdez, M.; Campoy, C. The role of probiotics and prebiotics in the prevention and treatment of obesity. Nutrients, 2019. 11(3), p.635. In line 357, what does ‘metabolized pattern’ mean? Is pattern of metabolites more appropriate?

Changes were effected in the manuscript as follows Line (507-509)

Studies on intestinal microbial metabolomes undoubtedly have considerable potential to aid in the determination of the metabolic pathway of gut microbiota and microbiota–host interactions.

In line 379, ‘questionable studies’ do not seem to be good words and can be insulting to the authors. Potential bad effects of “probiotics” need to be mentioned.

Changes were effected in the manuscript as follows Line (507-509)

Replace the word in this sentence uncertain

While the roles of microflora are currently unknown, several significant molecular mechanisms have been identified in a number of uncertain studies on bile salts, SCFAs, metabolic endotoxemia, and obesity.

Important papers regarding Akkermansia seem to be missing in the review

As per the reviewers recommendation, the reference paper regarding Akkermansia muciniphila were incorporated in the manuscript

Shin N-R et al. An increase in the Akkermansia sp. population induced by metformin treatment improves glucose homeostasis in diet-induced obese mice. Gut 63:727-735, 2014; Depommier C et al. Nat Med 25:1096, 2019; Barcena C et al. Healthspan and lifespan extension by fecal microbiota transplantation into progeroid mice ---. Nat Med in press, 2019.

References

Shin, N.R.; Lee, J.C.; Lee, H.Y.; Kim, M.S.; Whon, T.W.; Lee, M.S.; Bae, J.W. An increase in the Akkermansia spp. population induced by metformin treatment improves glucose homeostasis in diet-induced obese mice. Gut 2014; 63:727–35. Barcena, C.; Valdés-Mas, R.; Mayoral, P.; Garabaya, C.; Durand, S.; Rodríguez, F.; Fernández-García, M.T.; Salazar, N.; Nogacka, A.M.; Garatachea, N.; Bossut, N. Healthspan and lifespan extension by fecal microbiota transplantation into progeroid mice. Nat. Med, 2019. 25(8).1234-1242. Depommier, C.; Everard, A.; Druart, C.; Plovier, H.; Van Hul, M.; Vieira-Silva, S.; Falony, G.; Raes, J.; Maiter, D.; Delzenne, N.M.; de Barsy, M. Supplementation with Akkermansia muciniphila in overweight and obese human volunteers: A proof-of-concept exploratory study. Nat. Med, 2019. 25(7), p.1096.

Reviewer 2 Report

SUGGESTIONS:

Please cite the work of Dr Jeff Gordon who was the pioneer in establishing the role of the gut microbiome in obesity. Include the following references:

Backhead F. et al. PNAS 2004. vol 101(44):15718-15723 Peter J Turnbaugh et al. Nature 2009. vol 457: 480-484 RIdaura et al. 2013. Science Sept 6: 341(6150) Ruth Ley et al. 2011. Nature Reviews Microbiology 9(4):279-90 Takeshi Yoneshiro, Qiang Wang, Shingo Kajimura. BCAA catabolism in brown fat controls energy homeostasis through SLC25A44. Nature 2019.

Obesity in Kidney patients

Resolved: Being fat is good for Dialysis Patients: The Godzilla effect. J Am Soc Nephrol 19:1059-1064, 2008 ‘Obesity Paradox’: Patients with ESRD, obesity may have reduced risk for mortality Cohen J. A growing problem: Obesity in CKD and transplantation. Presented at NKF Spring Clinical Meetings; May 8-12, 2019.Boston

In addition it will be helpful to include work on the influence of high fat diet on immunoglobulin and immune response. Also needed is the effect of maternal obesity on the fetus. The effect of prebiotics and dietary fibers on obesity is equally important to include.

Author Response

Response to Reviewer 2 Comments

Manuscript ID: microorganisms-570580

Type of manuscript: Review

Title: Gut Microbiome Modulation based on Probiotic Application towards Anti-obesity: A Review on Efficacy and Validation

The authors were grateful for the editorial and reviewer’s valuable review comments to improvise the manuscript quality towards reader. The manuscript was revised according to the valuable suggestion. All changes in the revised manuscript were highlighted in red font. The language of the research article were edited by a MDPI English editing service (english-10989) and proof read. A point-by-point response to comments is included below. We are grateful to the reviewer for providing valuable comments on the article and also for your precious time.

SUGGESTIONS:

Please cite the work of Dr Jeff Gordon who was the pioneer in establishing the role of the gut microbiome in obesity. Include the following references: Backhead F. et al. PNAS 2004. vol 101(44):15718-15723 Peter J Turnbaugh et al. Nature 2009. vol 457: 480-484 RIdaura et al. 2013. Science Sept 6: 341(6150) Ruth Ley et al. 2011. Nature Reviews Microbiology 9(4):279-90 Takeshi Yoneshiro, Qiang Wang, Shingo Kajimura. BCAA catabolism in brown fat controls energy homeostasis through SLC25A44. Nature 2019.

As per the editors valuable suggestion the above mentioned references were update in the manuscript (Line-973 to 984)

Reference

Stenvinkel, P. Obesity in Kidney Disease. In: Rhee C., Kalantar-Zadeh K., Brent G. (eds) Endocrine Disorders in Kidney Disease. Springer, Cham, 2019.

187.Backhed, F.; Ding, H.; Wang, T.; Hooper, L.V.; Koh, G.Y.; Nagy, A.; Semenkovich, C.F.; Gordon, J.I. The gut microbiota as an environmental factor that regulates fat storage. PNAS. 2004. 101(44), pp.15718-15723.

188.Turnbaugh, P.J.; Hamady, M.; Yatsunenko, T.; Cantarel, B.L.; Duncan, A.; Ley, R.E.; Sogin, M.L.; Jones, W.J;, Roe, B.A.; Affourtit, J.P.; Egholm, M. A core gut microbiome in obese and lean twins. Nature, 2009. 457(7228), p.480.

189.Ridaura, V.K.; Faith, J.J.; Rey, F.E.; Cheng, J.; Duncan, A.E.; Kau, A.L.; Griffin, N.W.; Lombard, V.; Henrissat, B.; Bain, J.R.; Muehlbauer, M.J. Gut microbiota from twins discordant for obesity modulate metabolism in mice. Science, 2013. 41(6150), p.1241214.

Spor, A.; Koren, O.; Ley, R. Unravelling the effects of the environment and host genotype on the gut microbiome. Nat. Rev. Microbiol, 2011. 9(4), p.279. Obesity in Kidney patients Resolved: Being fat is good for Dialysis Patients: The Godzilla effect. J Am Soc Nephrol 19:1059-1064, 2008 Obesity Paradox’: Patients with ESRD, obesity may have reduced risk for mortality Cohen J. A growing problem: Obesity in CKD and transplantation. Presented at NKF Spring Clinical Meetings; May 8-12, 2019.Boston

In addition as per the reviewer recommendation, the obesity associated with kidney patients case study were incorporated in the manuscript (Line -249 to 280) 

Obesity associate other factor

Effect of probiotic in obesity associated kidney patients

Obesity upturns the possibility of CKD and the progression to the end-stage renal disease (ESRD). Probiotics could reduce the production of some uremic toxins. The protein metabolite hypothesis was proposed by Niwa et al. (112), which states that the organic anion transporters in the renal tubules take on toxins generated by protein putrefaction by the gut microbe. For example, Lactobacillus murinus supplementation prevents hypertension from developing high-salt intake in hypertensive mice (113,114). Ichii et al. (115) showed that LPS exhibited mouse podocytes displayed pro-inflammatory phenotype, reduced podocyte-specific gene expression, and reduced cell viability. In the case of end-stage kidney disease (ESRD) patients, approximately 190 microbial operational taxonomic units were considered distinct in abundance compared with healthy checks (116). In CKD patients, the lower numbers of Prevotellaceae and Lactobacillaceae families (both regarded to be natural colonic microbiota) Enterococci and Enterobacteria species 100 times greater were identified (116). Likewise, an oral supplement to the MLR mice with spontaneous lupus nephritis enhanced kidney function and general survival by decreasing the intestinal permeability and systemic inflammation of the symbiont strains of Lactobacillus (Lactobacillus gasseri, Lactobacillus rhamnosus, Lactobacillus oris, Lactobacillus reuteri, and Lactobacillus johnsonii). In another study found an enhanced gastrointestinal barrier, reduced internal swelling and accumulation of uremic toxins in spontaneous 5/6 hypertensive nephrectomized rats (a typical CKD model) treated with Lactobacillus acidophilus, resulting in a reduction in kidney damage (117). Treatment with various bacteria has lowered lipid peroxidation and improved antioxidant enzymes such as superoxide dismutase (SOD) and catalase (CAT) in acetaminophen-induced uremic rats and chrome-induced oxidative stress rats (118-120). Also, reduced oxidative stress was correlated with reduced renal necrosis (119, 121,122).

In summary, probiotics can protect renal injury and dysfunction by reducing swelling, apoptosis, and oxidative stress. Some findings might be inconsistent because of the distinction between bacterial species and the patient population or animal models used during the research. The concept, indeed, that obesity can change the intestinal environment through bacterial composition can be considered as a cause of renal injury. Notable findings indicate that particular personal and microbiome characteristics allow specific glucose response predictions, which can result in customized microbiome modulation through dietary, prebiotic and probiotic intervention (186). Modulation of gut's bacterial balance for using probiotics could be a suitable choice in cases of obesity to improve the kidney function.

References

Niwa, T.; Takeda, N.; Tatematsu, A.; Maeda, K. Accumulation of indoxyl sulfate, an inhibitor of drug-binding, in uremic serum as demonstrated by internal-surface reversed-phase liquid chromatography. Clin. Chem. 1988. 34(11), pp.2264-2267.

113.Oh, M.S.; Phelps, K.R.; Traube, M.; Barbosa-Saldivar, J.L.; Boxhill, C.; Carroll, H.J. D-lactic acidosis in a man with the short-bowel syndrome. N Engl J Med, 1979. 301(5), pp.249-252.

114.Tang, W.W.; Wang, Z.; Levison, B.S.; Koeth, R.A.; Britt, E.B.; Fu, X.; Wu, Y.; Hazen, S.L. 2013. Intestinal microbial metabolism of phosphatidylcholine and cardiovascular risk. N Engl J Med, 2013. 368(17), pp.1575-1584.

Ichii, O.; Otsuka-Kanazawa, S.; Nakamura, T.; Ueno, M., Kon, Y.; Chen, W.; Rosenberg, A.Z.; Kopp, J.B. Podocyte injury caused by indoxyl sulfate, a uremic toxin and aryl-hydrocarbon receptor ligand. PLoS One, 2014. 9(9), p.e108448.

116.Vaziri, N.D.; Wong, J.; Pahl, M.; Piceno, Y.M.; Yuan, J.; DeSantis, T.Z.; Ni, Z.; Nguyen, T.H.; Andersen, G.L. Chronic kidney disease alters intestinal microbial flora. Kidney Int, 2013. 83(2), pp.308-315.

117.Zhao, Y.Y.; Wang, H.L.; Cheng, X.L.; Wei, F.; Bai, X.; Lin, R.C.; Vaziri, N.D. 2015. Metabolomics analysis reveals the association between lipid abnormalities and oxidative stress, inflammation, fibrosis, and Nrf2 dysfunction in aristolochic acid-induced nephropathy. Sci. Rep, 2015. 5, p.12936.

118.Chen, D.Q.; Chen, H.; Chen, L; Vaziri, N.D.; Wang, M.; Li, X.R.; Zhao, Y.Y. The link between phenotype and fatty acid metabolism in advanced chronic kidney disease. Nephrol Dial Transplant. 2017;32:1154–66.

Wang, Z.; Koonen, D.; Hofker, M.; Fu, J.Y. Gut microbiome and lipid metabolism: from associations to mechanisms. Curr Opin Lipidol. 2016;27:216–24.

120.Xu, K.Y.; Xia, G.H.; Lu, J.Q.;, Chen, M.X.; Zhen, X.; Wang, S.; You, C.; Nie, J.; Zhou, H.W.; Yin, J. Impaired renal function and dysbiosis of gut microbiota contribute to increased trimethylamine-N-oxide in chronic kidney disease patients. Sci Rep. 2017;7:1445.

Vaziri, N.D.; Yuan, J.; Nazertehrani, S.; Ni, Z.; Liu, S. Chronic kidney disease causes disruption of gastric and small intestinal epithelial tight junction. Am J Nephrol. 2013;38:99–103. Ikizler, T.A. Resolved: being fat is good for dialysis patients: the Godzilla effect: pro. J Am Soc Nephrol. 2008 Jun;19(6):1059-62. doi:10.1681/ASN.2007090983. Epub 2008 Feb 6. PubMed PMID: 18256355.

In addition it will be helpful to include work on the influence of high fat diet on immunoglobulin and immune response.

As per the editors valuable suggestion the relation between high fat diet and immune response were briefed with suitable reference were update in the manuscript (Line-314 to 338)

Correlation of obesity with immune system and transmission through next generation

Influences of high fat diet in obesity patients in correlation immunosenescence

Increased intestinal mucosal absorption (e.g. leaky gut syndrome), especially in relation to immune modifications, can lead to important harm to the GI tract, causing bacteria, toxins, nutrients, and waste to leak from the bowels into the bloodstream, causing a severe/autoimmune response, in particular, if such toxins deplete the liver (133). In the mouse model, Moya-Pérez et al. (134) found gut ecosystem with B. pseudocatenulatum CECT 7765 modulates HFD-induced infiltration of the immune cells, intestinal and peripheral inflammation, respectively improvements in obesity-related metabolic dysfunction. The study also shows anti-inflammatory effects of these bifidobacterial strains involve both inborn and adaptive immune functions involving B lymphocytes. Zhang et al. (135), two strains of Lactobacillus rheuteri were piglet-insulated as ZJ617, with strong adhesive capabilities, and ZJ615 with low adhesive capacities. Likewise, this researcher analyzed the immunomodulatory effects of strains with diverse adhesive abilities. Kowalska et al. (136) were first outlined, and numerous trials still show that leptin is elevated under the impact of HFD relative to control diets in rats (137-139). It has been proved in rats using the same diet that HFD increases leptin content (140). In rats, leptin might be reduced with HFD within 72 hours, regardless of body weight rise (141). The reasons for the immunosenescence phenomenon have not yet been fully explained. In addition, there have been several confounders such as body fat mass which separately influence the immune response, but which can differ significantly between the heterogeneous aging people. An elevated body fat mass was suggested to not have the same adverse effect on the elderly as for a young population. This was concluded with research showing a survival benefit for geriatric populations above the age of 65 when the BMI reaches 25 kg / m2 generally identified as an overweight (142,143). In addition to fat tissue and integrated immune cells, the mediator is known to be immediately involved, such as adipocytes or several mostly pro-inflammatory cytokines (143-145), which migrate to the immune function. Nevertheless, the dietary intake of probiotics has widely considered health-effective, mainly due to its immunomodulatory properties in obesity treatment.

References

Sharma, R.; Kapila, R.; Kapila, S. Probiotics as anti-immunosenescence agents. Food Rev. Int, 2013. 29(2), 201-216. Moya-Pérez, A.; Neef, A.; Sanz, Y. Bifidobacterium pseudocatenulatum CECT 7765 reduces obesity-associated inflammation by restoring the lymphocyte-macrophage balance and gut microbiota structure in high-fat diet-fed mice. PLoS One, 2015. 10(7), p.e0126976.

135.Zhang, W.; Wang, H.; Liu, J.; Zhao, Y.; Gao, K.; Zhang, J. Adhesive ability means inhibition activities for lactobacillus against pathogens and S-layer protein plays an important role in adhesion. Anaerobe, 2013. 22, pp.97-103.

Kowalska, I.; Straczkowski, M.; Gorski, J.; Kinalska, I. The effect of fasting and physical exercise on plasma leptin concentrations in high-fat fed rats. J Physiol Pharmacol. 1999; 50:309–20.

137.Gan, L.; England, E.; Yang, J.Y.; Toulme, N.; Ambati, S.; Hartzell, D.L. A 72-hour high fat diet increases transcript levels of the neuropeptide galanin in the dorsal hippocampus of the rat. BMC Neurosci. 2015; 16:51.

138.Yan, W.J.; Mu, Y.; Yu, N.; Yi, T.L.; Zhang, Y.; Pang, X.L. Protective effects of metformin on reproductive function in obese male rats induced by high-fat diet. J Assist Reprod Genet. 2015; 32:1097–104.

Abu, M.N.; Samat, S.; Kamarapani, N.; Nor, H.F.; Wan Ismail, W.I.; Hassan, H.F. Tinospora crispa Ameliorates Insulin Resistance Induced by High Fat Diet in Wistar Rats. Evid Based Complement Alternat Med. 2015; 2015:985042.

140.Zhou, L.; Jang, K.Y.; Moon, Y.J.; Wagle, S.; Kim, K.M.; Lee, K.B. Leptin ameliorates ischemic necrosis of the femoral head in rats with obesity induced by a high-fat diet. Sci Rep. 2015. 5:9397

141.Bollheimer, L.C.; Buettner, R.; Pongratz, G.; Brunner-Ploss, R.; Hechtl, C.; Banas. Sarcopenia in the ageing high-fat fed rat: a pilot study for modeling sarcopenic obesity in rodents. Biogerontology. 2012; 13:609–20.

Faragher, R.; Frasca, D.; Remarque, E.; Pawelec, G. Better immunity in later life: a position paper. Age (Dordr). 2014; 36(3):9619. Iweala, O.I.; Nagler, C.R. The Microbiome and Food Allergy. Annu. Rev. Immunol, 2019. 37, pp.377-403. Yousefi, B.; Eslami, M.; Ghasemian, A.; Kokhaei, P.; Sadeghnejhad, A. Probiotics can really cure an autoimmune disease?. Gene Reports, 2019. 100364. Childs, C.E.; Calder, P.C.; Miles, E.A. Diet and Immune Function. Nutrients. 2019, 11, 1933.

Also needed is the effect of maternal obesity on the fetus.

As per the editors valuable suggestion the relation between maternal obesity and fetus were briefed with suitable reference were update in the manuscript (Line- 339 to 371)

Effect of Maternal Obesity towards new born fetus

In recent decades the incidence of several prenatal and early post-natal variables connected with the growth of childhood adiposity (such as prematurity and low birth weight (146-148), gestational diabetes (149), surplus body mass gain during gestation or child formula feeding has also been unbroken in recent decades, in conjunction with enhanced prevalence of childhood obesity. Interestingly, the incidence of these perinatal risk factors has risen more firmly in developing countries than in developing countries (150-152). Maternal diabetes mellitus (GDM) increases the risk of disproportionate adiposity and macrosomia in infants. Long-term maternal GDM is correlated with a greater danger of child obesity and baby metabolism (153, 154). Boyle et al. (155), revealed parental obesity raises the capability of mesenchymal umbilical cord stems for abiogenesis, which leads to infant adiposity. Childhood obesity is of significant concern, given the elevated danger that obese kids will become obese adults and thus develop severe comorbidities such as metabolic syndrome, diabetes, and cardiovascular disorders (156). The gut microbiome is acknowledged as an environmental factor in recent innovations in gnotobiotic mice sequencing technology that influences the metabolism of the entire organism by influence the energy equilibrium along with regulatory signals of peripheral and essential food intakes, inflammation, and intestinal block functions, thus stimulating body weight. Gut microbiota is a specific entity in the body which has its own genome with a gene pool bigger than its host. A novel relationship between obesity and diabetes has been ascribed to the extensive physiological functions of intestinal gut microbiota in extra-intestinal tissues, like adipose tissue (157,158). It thus plays a significant role in obesity and diabetes disease pathology. Based on studies with germ-free mice that have shown protection from dietary-inducing obesity (DIO) growth, the underlying processes of the microbiota attributable to host metabolism were first shown (153, 156, 158). Notable animal studies have concluded that probiotics during pregnancy and lactation reduced maternal HFD induced dietary programming associated with maternal obesity, which suggests that modified parental gut microbiota might be a helpful approach for enhancing paternal and offspring metabolic findings (159). Vähämiko et al. (161), human studies reveal that probiotic supplements throughout pregnancy may impact the DNA methylation status by some obesity promoters and weight gain genes in both mothers and their offspring. Everard et al. (160), exposed that L. rhamnosus GG started one month before birth and persisted until the six months after birth altered children's development pattern, by reducing unjustified weight gains during the infancy years of a child. Promising prevention and therapeutic approach for GDM may be the use of probiotic supplementation. Microbiota dysbiosis was supplemented by probiotics and probiotics emerge as an efficient intervention to improve the entire health of the preterm infants.

References

146.Yuan, Z. P.; Yang, M.; Liang, L.; Fu, J. F.; Xiong, F.; Liu, G. L.; Zhang, S. Possible role of birth weight on general and central obesity in Chinese children and adolescents: a cross-sectional study. Ann. Epidemiol., 2015, 25.10: 748-752.

Rogers, I. The influence of birth weight and intrauterine environment on adiposity and fat distribution in later life. Int. J. Obes. Relat. Metab. Disord. 2003. 27, 755–777.

148.Rockenbach, G.; Luft, V. C.; Mueller, N. T.; Duncan, B. B.; Stein, M. C.; Vigo, Á.; Appel, L. J. Sex-specific associations of birth weight with measures of adiposity in mid-to-late adulthood: the Brazilian Longitudinal Study of Adult Health (ELSA-Brasil). IJO. 2016. 40(8), 1286.

149.Logan, K. M.; Gale, C.; Hyde, M. J.; Santhakumaran, S.; Modi, N. Diabetes in pregnancy and infant adiposity: systematic review and meta-analysis. Arch. Dis. Child Fetal Neonatal Ed. 2017. 102, F65–F72.

150.Blencowe, H.; Cousens, S.; Oestergaard, M. Z.; Chou, D.; Moller, A. B.; Narwal, R.; Lawn, J. E. National, regional, and worldwide estimates of preterm birth rates in the year 2010 with time trends since 1990 for selected countries: a systematic analysis and implications. The lancet, 2012. 379(9832), 2162-2172.

Harrison, M. S.; Goldenberg, R. L. Global burden of prematurity. Semin. Fetal Neonatal Med. 2016. 21, 74–79. Lee, A. C. et al. Estimates of burden and consequences of infants born small for gestational age in low and middle income countries with INTERGROWTH-21st standard: analysis of CHERG datasets. BMJ. 2017. 358, j3677 (2017).

153.Crume, T.L.; Ogden, L.; West, N.A.; Vehik, K.S.; Scherzinger, A.; Daniels, S.; McDuffie, R.; Bischoff, K.; Hamman, R.F.; Norris, J.M.; Dabelea, D. Association of exposure to diabetes in utero with adiposity and fat distribution in a multiethnic population of youth: the Exploring Perinatal Outcomes among Children (EPOCH) Study. Diabetologia 2011; 54:87–92.

Malcolm, J. Through the looking glass: gestational diabetes as a predictor of maternal and offspring long-term health. Diabetes Metab Res Rev 2012; 28:307–11.

155.Boyle, K.E.; Patinkin, Z.W.; Shapiro, A.L.B.; Baker, I.I.P.R.; Dabelea, D.; Friedman, J.E. Mesenchymal stem cells from infants born to obese mothers exhibit greater potential for adipogenesis: The Healthy Start Baby BUMP Project. Diabetes 2016; 65:647–59.

156.Whitaker, R.C.; Wright, J.A.; Pepe, M.S.; Seidel, K.D.; Dietz, W.H. Predicting obesity in young adulthood from childhood and parental obesity. N Engl J Med 1997; 337:869–73.

157.Bäckhed, F.; Ding, H.; Wang, T.; Hooper, L.V.; Koh, G.Y.; Nagy, A.; Semenkovich, C.F.; Gordon, J.I. The gut microbiota as an environmental factor that regulates fat storage. PNAS USA, 2004 101(44), pp.15718-15723.

Luoto, R.; Kalliomaki, M.; Laitinen, K.; Isolauri, E. The impact of perinatal probiotic intervention on the development of overweight and obesity: follow-up study from birth to 10 years. Int J Obes (Lond) 2010; 34:1531–7.

159.Shin, N.R.; Lee, J.C.; Lee, H.Y.; Kim, M.S.; Whon, T.W.; Lee, M.S.; Bae, J.W. An increase in the Akkermansia spp. population induced by metformin treatment improves glucose homeostasis in diet-induced obese mice. Gut 2014; 63:727–35.

160.Everard, A.; Belzer, C.; Geurts, L.; Ouwerkerk, J.P.; Druart, C.; Bindels, L.B.; Guiot, Y.; Derrien, M.; Muccioli, G.G.; Delzenne, N.M.; de Vos, W.M.; Cani, P.D. Cross-talk between Akkermansia muciniphila and intestinal epithelium controls diet-induced obesity. Proc Natl Acad Sci U S A, 2013; 110:9066–71.

161.Vähämiko, S.; Laiho, A.; Lund, R.; Isolauri, E.; Salminen, S.; Laitinen, K. The impact of probiotic supplementation during pregnancy on DNA methylation of obesity-related genes in mothers and their children. Eur. J. Nutr. 2018.  doi: 10.1007/s00394-017-1601-1

The effect of prebiotics and dietary fibers on obesity is equally important to include.

As per the editors valuable suggestion the relation between maternal obesity and fetus were briefed with suitable reference were update in the manuscript (Line- 281 to 312)

Effect of Prebiotic and dietary fiber towards obesity treatment

FAO / WHO described prebiotics as "non-digestible food" products that benefit the host by selectively promoting development and activities of one or more of the existing colon bacterial species, thus improving host health (123). This perception usually includes in-digestible, non- hydrolyzable forms of carbohydrate (e.g. fructooligosaccharides (FOS), galactooligosaccharidess (GOSs), soybean oligosaccharides, cyclodextrins, inulin, gluco-oligosaccharides, xylooligosaccharides, lactulose, lacto-sucrose, and isomaltooligosaccharides) which have the potential to reach the distal parts of the human gastrointestinal tract (124). Increasing evidence that prebiotic therapy positively influences the composition of gut microbes, enhance the growth of Lactobacillus and Bifidobacterium in the gastrointestinal tract of obese animals (125). For instance, in human breast milk is a wealthy source of human milk oligosaccharides (prebiotics candidate), which boost the development of beneficial bacteria (Bacteroides and Bifidobacterium) and inhibit the adherence of pathogens like E. coli, Campylobacter jejuni and Hélicobacter pylori (128). Some trials have shown these modifications in the oligofructose-treated obesity and diabetic mice (126) as much as the prebiotic carbohydrate-treated ob/ob mice have to do with accelerated entero-endocrine cell growth, glucose homeostatic and leptin sensitivity (127). These changes were also related with high production of intracellular glucagon-like peptide-2 (GLP-2), the intestinotrophic pro-glucagon-derived peptide associated in gut's permeability, thus decreasing both obesity-related gastric inflammatory and cardiovascular disorders. In this context, Everard et al. (126) stated on non-obese metabolic phenotypes described by decreased triglyceride concentrations, tissue adipose, and muscle lipid infiltration in the oligofructose-treated animal.  Widely proof of positive impacts of L. rhamnosus GG therapy in the pediatric population regarding obesity-related non-alcoholic liver fatty diseases (NAFLD). In most clinical research the hepatic accumulation of TAG and/or cholesterol in the liver tissue, as specified as steatosis, can be reduced by prebiotics. This impact might be interesting, as 25 to 75% of obese people have non-alcoholic fatty liver disease. Although the energy intake function is concerned, certain studies have not shown any effects on the long-term prebiotic supplement in an of pre-meal inulin and galacto-oligosaccharides or on the short-term fructose- oligosaccharides (129) treatments; others have shown that oligofructose or inulin intake in both non-obese and obese individuals have decreased their overall energy intakes for at least two weeks (129-132). Several findings of prebiotics showed significant beneficial outcomes on the body weight, waist circumference, BMI, lipid profile, fat deposition, and chronic inflammation status of prebiotics, and these may lead to alternative approaches to management and treatment of obesity and associated metabolic disorders (186).

References

123.Pineiro, M.; Asp, N.-G.; Reid, G.; Macfarlane, S.; Morelli, L.; Brunser, O.; Tuohy, K. FAO Technical meeting on prebiotics. J. Clin. Gastroenterol. 2008, 42, S156–S159.

Younis, K.; Ahmad, S.; Jahan, K. Health benefits and application of prebiotics in foods. J. Food Process. Technol. 2015, 6, 1. Connolly, M.L.; Lovegrove, J.A.; Tuohy, K.M. In vitro fermentation characteristics of whole grain wheat flakes and the effect of toasting on prebiotic potential. J. Med. Food 2012, 15, 33–43.

126.Everard, A.; Lazarevic, V.; Derrien, M.; Girard, M.; Muccioli, G.M.; Neyrinck, A.M.; Possemiers, S.; Van Holle, A.; François, P.; de Vos,W.M.; et al. Responses of gut microbiota and glucose and lipid metabolism to prebiotics in genetic obese and diet-induced leptin-resistant mice. Diabetes 2011, 60, 2775–2786.

127.Cani, P.D.; Possemiers, S.; Van de Wiele, T.; Guiot, Y.; Everard, A.; Rottier, O.; Geurts, L.; Naslain, D.; Neyrinck, A.; Lambert, D.M.; et al. Changes in gut microbiota control inflammation in obese mice through a mechanism involving GLP-2-driven improvement of gut permeability. Gut 2009, 58, 1091–1103.

Newburg, D. S. Oligosaccharides in human milk and bacterial colonization. J Pediatr Gastroenterol Nutr, 2000. 30, S8-S17. Cerdó, T.; García-Santos, J.A.; G Bermúdez, M.; Campoy, C. The role of probiotics and prebiotics in the prevention and treatment of obesity. Nutrients, 2019. 11(3), p.635.

130.Sanders, M.E.; Merenstein, D.J.; Reid, G.; Gibson, G.R.; Rastall, R.A. Probiotics and prebiotics in intestinal health and disease: from biology to the clinic. Nat. Rev. Gastroenterol. Hepatol. 2019. p.1.

131.Mishra, S.P.; Wang, S.; Nagpal, R.; Miller, B.; Singh, R.; Taraphder, S.; Yadav, H. Probiotics and prebiotics for the amelioration of type 1 diabetes: Present and future perspectives. Microorganisms, 2019. 7(3), p.67.

Hadi, A.; Mohammadi, H.; Miraghajani, M.; Ghaedi, E. Efficacy of synbiotic supplementation in patients with nonalcoholic fatty liver disease: a systematic review and meta-analysis of clinical trials: synbiotic supplementation and NAFLD. Crit. Rev. Food Sci. Nutr, 2019. 59(15), pp.2494-2505. Sharma, R.; Kapila, R.; Kapila, S. Probiotics as anti-immunosenescence agents. Food Rev. Int, 2013. 29(2), 201-216. Moya-Pérez, A.; Neef, A.; Sanz, Y. Bifidobacterium pseudocatenulatum CECT 7765 reduces obesity-associated inflammation by restoring the lymphocyte-macrophage balance and gut microbiota structure in high-fat diet-fed mice. PLoS One, 2015. 10(7), p.e0126976.

135.Zhang, W.; Wang, H.; Liu, J.; Zhao, Y.; Gao, K.; Zhang, J. Adhesive ability means inhibition activities for lactobacillus against pathogens and S-layer protein plays an important role in adhesion. Anaerobe, 2013. 22, pp.97-103.

Kowalska, I.; Straczkowski, M.; Gorski, J.; Kinalska, I. The effect of fasting and physical exercise on plasma leptin concentrations in high-fat fed rats. J Physiol Pharmacol. 1999; 50:309–20.

137.Gan, L.; England, E.; Yang, J.Y.; Toulme, N.; Ambati, S.; Hartzell, D.L. A 72-hour high fat diet increases transcript levels of the neuropeptide galanin in the dorsal hippocampus of the rat. BMC Neurosci. 2015; 16:51.

138.Yan, W.J.; Mu, Y.; Yu, N.; Yi, T.L.; Zhang, Y.; Pang, X.L. Protective effects of metformin on reproductive function in obese male rats induced by high-fat diet. J Assist Reprod Genet. 2015; 32:1097–104.

Abu, M.N.; Samat, S.; Kamarapani, N.; Nor, H.F.; Wan Ismail, W.I.; Hassan, H.F. Tinospora crispa Ameliorates Insulin Resistance Induced by High Fat Diet in Wistar Rats. Evid Based Complement Alternat Med. 2015; 2015:985042.

140.Zhou, L.; Jang, K.Y.; Moon, Y.J.; Wagle, S.; Kim, K.M.; Lee, K.B. Leptin ameliorates ischemic necrosis of the femoral head in rats with obesity induced by a high-fat diet. Sci Rep. 2015. 5:9397

141.Bollheimer, L.C.; Buettner, R.; Pongratz, G.; Brunner-Ploss, R.; Hechtl, C.; Banas. Sarcopenia in the ageing high-fat fed rat: a pilot study for modeling sarcopenic obesity in rodents. Biogerontology. 2012; 13:609–20.

Faragher, R.; Frasca, D.; Remarque, E.; Pawelec, G. Better immunity in later life: a position paper. Age (Dordr). 2014; 36(3):9619. Iweala, O.I.; Nagler, C.R. The Microbiome and Food Allergy. Annu. Rev. Immunol, 2019. 37, pp.377-403. Yousefi, B.; Eslami, M.; Ghasemian, A.; Kokhaei, P.; Sadeghnejhad, A. Probiotics can really cure an autoimmune disease?. Gene Reports, 2019. 100364. Childs, C.E.; Calder, P.C.; Miles, E.A. Diet and Immune Function. Nutrients. 2019, 11, 1933.

Round 2

Reviewer 1 Report

This paper has been improved by the incorporation of the reviewers’ comments. However, still several questions remain to be addressed.

In lines 131-132, it is hard to understand what the sentence ‘The glycine or taurine derived from cholesterol produced from in the Liver” means? Does it mean glycine or taurine liberated from cholesterol-glycine conjugate? In lines 167-168, it is hard to understand what the sentence ‘metabolic endotexemia was observed in genetically obese mice (C57Bl6/j) consume normal chow and mackerel based chow that consume the obesogenic diet” means? In line 170, HFD-fed mice could be correct rather than HFD mice. In line 172, LPS binding CD14 gene does not activate NF-kB. In line 171, where (in which tissue) do the HFD-fed mice show a greater abundance of LPS-containing bacteria? Please specify. In line 234, do author mean “metabolites produced by some bacteria”? Bacteria are not signals. In line 389, ‘such that’ could be better than ‘such as’. In line 409, do authors mean ‘fish oil-rich diet increases the proportion of Actinobacteria’? In line 435, it is hard to understand what the sentence ‘L. rhamnosus GG can replace HFD” means.

10. Lines 472-473 are still hard to understand.

Lines 508-511 are still awkward and the words such as ‘strange’ or ‘uncertain’ could be insulting to the authors of the papers

Author Response

Manuscript ID: microorganisms-570580

Type of manuscript: Review

Title: Gut Microbiome Modulation Based on Probiotic Application for Anti-obesity: A Review on Efficacy and Validation

The authors were grateful for the editorial and reviewer’s valuable comments to improvise the manuscript quality towards reader. The manuscript was revised according to the valuable suggestion. All the changes in the revised manuscript were highlighted. The language of the review article were edited through MDPI English editing service (English-12750) and proof read by native english speakers. A point-by-point response to comments is included below. We are grateful to the reviewer for providing valuable comments on the article and also for your precious time.

This paper has been improved by the incorporation of the reviewers’ comments. However, still several questions remain to be addressed.

In lines 131-132, it is hard to understand what the sentence ‘The glycine or taurine derived from cholesterol produced from in the Liver” means? Does it mean glycine or taurine liberated from cholesterol-glycine conjugate? In lines

Changes were effected in the manuscript (Line: 134-135)

Bile acids actively help to resolve dietary fat uptake in the small intestine. The glycine or taurine conjugated with cholesterol, which are synthesized in the Liver.

167-168, it is hard to understand what the sentence ‘metabolic endotexemia was observed in genetically obese mice (C57Bl6/j) consume normal chow and mackerel based chow that consume the obesogenic diet” means?

Changes were effected in the manuscript (Line: 170-176)

Metabolic endotoxemia (Induce pro-inflammatory and oxidative stress) was observed in obese mice (C57bl6/J), which consume normal chow (composed of ground wheat, corn, or oats, alfalfa and soybean meals, a protein with minerals and vitamins) but it might be induced in normal mice through obesogenic diet (high fat diet).

In line 170, HFD-fed mice could be correct rather than HFD mice.

Changes were effected in the manuscript (Line: 174)

As per reviewer’s valuable recommendation we had edited the respective word and rephrased as follows 

In a comparison, the proportional abundance of Firmicutes was significantly lower in low-fat diet (LFD) than high-fat diet (HFD) fed mice, but Proteobacteria were not significantly changed between the two groups [80].

References

Dalby, M. J.; Aviello, G.; Ross, A. W.; Walker, A. W.; Barrett, P., & Morgan, P. J. Diet induced obesity is independent of metabolic endotoxemia and TLR4 signalling, but markedly increases hypothalamic expression of the acute phase protein, SerpinA3N. Sci. Rep, 2018, 8(1), 15648. doi:10.1038/s41598-018-33928-4. In line 172, LPS binding CD14 gene does not activate NF-kB.

The specific sentence were deleted during editing process.

In line 171, where (in which tissue) do the HFD-fed mice show a greater abundance of LPS-containing bacteria? Please specify.

Changes were effected in the manuscript (Line: 170-176)

Metabolic endotoxemia (Induce pro-inflammatory and oxidative stress) was observed in obese mice (C57bl6/J), which consume normal chow (composed of ground wheat, corn, or oats, alfalfa and soybean meals, a protein with minerals and vitamins) but it might be induced in normal mice through obesogenic diet (high fat diet). Comparing the proportional abundance of Firmicutes was significantly lower in LFD than HFD fed mice but Proteobacteria were not significantly changed between HFD and LFD fed mice [80].

References

Dalby, M. J.; Aviello, G.; Ross, A. W.; Walker, A. W.; Barrett, P., & Morgan, P. J. Diet induced obesity is independent of metabolic endotoxemia and TLR4 signalling, but markedly increases hypothalamic expression of the acute phase protein, SerpinA3N. Sci. Rep, 2018, 8(1), 15648. doi:10.1038/s41598-018-33928-4.

In line 234, do author mean “metabolites produced by some bacteria”? Bacteria are not signals.

Changes were effected in the manuscript (Line: 237-239)

Some of the Lactic acid bacteria can convert glutamate into gamma-aminobutyric acid (GABA), express GABA-binding proteins (102, 108) and serotonin (96, 101, 109, 110, 111), which act as a synaptic neurotransmitter signals that regulates appetite.

References

Ley, R.E.; Bäckhed, F.; Turnbaugh, P.; Lozupone, C.A.; Knight, R.D.; Gordon, J.I. Obesity alters gut microbial ecology. Proc Natl Acad Sci USA 2005; 102: 11070–75. Schellekens, H.; Dinan, T.G.; Cryan, J.F. Lean mean fat reducing “ghrelin” machine: hypothalamic ghrelin and ghrelin receptors as therapeutic targets in obesity. Neuropharmacology. 2010; 58: 2–16. Bodenlos, J.S.; Schneider, K.L.; Oleski, J.; Gordon, K.; Rothschild, A.J.; Pagoto, S.L. Vagus nerve stimulation and food intake: effect of body mass index. J Diabetes Sci Technol. 2014; 8: 590–95. Meng, F.; Han, Y.; Srisai, D. New inducible genetic method reveals critical roles of GABA in the control of feeding and metabolism. Proc Natl Acad Sci USA 2016; 113: 3645–50. Delgado, T.C. Glutamate and GABA in appetite regulation. Front Endocrinol (Lausanne) 2013; 4: 103. Heisler, L.K.; Jobst, E.E.; Sutton, G.M. Serotonin reciprocally regulates melanocortin neurons to modulate food intake. Neuron 2006; 51: 239–49. Xu, Y.; Jones, J.E.; Kohno, D.; Williams, K.W.; Lee, C.E.; Choi, M.J.; Anderson, J.G.; Heisler, L.K.; Zigman, J.M.; Lowell, B.B.; Elmquist, J.K. 5-HT2CRs expressed by pro-opiomelanocortin neurons regulate energy homeostasis. Neuron. 2008. 60(4):582-9.

In line 389, ‘such that’ could be better than ‘such as’.

Changes were effected in the manuscript (Line: 391)

As per reviewer’s valuable recommendation we had edited the respective word and rephrased as follows 

The impact of specific diet have an specific influence on enterotype change such that Bacteroides are associated with a rich protein and animal fat diet, while Prevotella is associated with high carbohydrate consumption.

In line 409, do authors mean ‘fish oil-rich diet increases the proportion of Actinobacteria’?

Changes were effected in the manuscript (Line: 407-410)

High-fat diets rich in both fish oil (ω3-PUFAs) and Pig (Lard) fat (HL diet, mainly saturated fatty acids) has a stronger impact on microbiota modulation , but pig fat induce stronger impact on Bacteroidetes and Bilophila enhancement, while fish oil induced increased Actinobacteria (169, 170).

References

Sonnenburg, E.D.; Sonnenburg, J.L. Starving our microbial self: the deleterious consequences of a diet deficient in microbiota-accessible carbohydrates. Cell Metab. 2014; 20(5):779–86. Desai, M.S.; Seekatz, A.M.; Koropatkin, N.M.; Kamada, N.; Hickey, C.A.; Wolter, M. Adietary fiber-deprived gut microbiota degrades the colonic mucus barrier and enhances pathogen susceptibility. Cell. 2016; 167(5). 1339-53.e21

In line 435, it is hard to understand what the sentence ‘L. rhamnosus GG can replace HFD” means.

Changes were effected in the manuscript (Line: 433-436)

rhamnosus GG treatment replace HFD through reduction of adiposity in high-fat diet-fed mice through enhancement of adiponectin production, which protect animals from insulin resistance and can reduce liver adiposity (173).

                  References

Kim, S.W.; Park, K.Y.; Kim, B.; Kim, E.; Hyun, C.K. Lactobacillus rhamnosus GG improves insulin sensitivity and reduces adiposity in high-fat diet-fed mice through enhancement of adiponectin production. Biochem Biophys Res Commun. 2013. 431(2):258–63.

Lines 472-473 are still hard to understand.

Changes were effected in the manuscript (Line: 471-472)

In contrast, it was shown that 92 % of Clostridium difficile infection (CDI) cases were eliminated through fecal microbiota transplantation (FMT) (176).

                 References

de Vos, W.M. Fame and future of faecal transplantations – developing next-generation therapies with synthetic microbiomes. Microb. Biotechnol. 2013. 6, 316–325.

Lines 508-511 are still awkward and the words such as ‘strange’ or ‘uncertain’ could be insulting to the authors of the papers 

Changes were effected in the manuscript (Line: 506-509)

The task of this review was to investigate the body of literature on the use of probiotics for obesity treatment. Probiotics showed weight reduction characteristics in different animal research, human studies, and several factors have been suggested to explain these anti-obesity effects. Previously the role of microflora towards metabolic regulation, disease, genetic disorder, were unexplored parts of gastroenterology. Currently the roles of microflora were identified through molecular sequencing such metagenomics and metabolomics and its significant co regulation were studied extensively on bile salts, SCFAs, metabolic endotoxemia, and obesity. The specificity of probiotics, however, makes it difficult to determine a specialized path for their operating mechanism.